# On Private and Robust Bandits

**Yulian Wu**[*]
KAUST
yulian.wu@kaust.edu.sa

**Xingyu Zhou**[*]
Wayne State University
xingyu.zhou@wayne.edu

**Youming Tao**
Shandong University
ym.tao99@mail.sdu.edu.cn

**Di Wang**
KAUST
di.wang@kaust.edu.sa

## Abstract

We study private and robust multi-armed bandits (MABs), where the agent receives Huber's contaminated heavy-tailed rewards and meanwhile needs to ensure differential privacy. We consider both the finite $k$-th raw moment and the finite $k$-th central moment settings for heavy-tailed rewards distributions with $k \geq 2$. We first present its minimax lower bound, characterizing the information-theoretic limit of regret with respect to privacy budget, contamination level, and heavy-tailedness. Then, we propose a meta-algorithm that builds on a private and robust mean estimation sub-routine PRM that essentially relies on reward truncation and the Laplace mechanism. For the above two different heavy-tailed settings, we give corresponding schemes of PRM, which enable us to achieve nearly-optimal regrets. Moreover, our two proposed truncation-based or histogram-based PRM schemes achieve the optimal trade-off between estimation accuracy, privacy and robustness. Finally, we support our theoretical results and show the effectiveness of our algorithms with experimental studies.

## 1 Introduction

The multi-armed bandit (MAB) [1] problem provides a fundamental framework for sequential decision-making under uncertainty with bandit feedback, which has drawn a wide range of applications in medicine [2], finance [3, 4], recommendation system [5], and online advertising [6], to name a few. Consider a portfolio selection in finance as an example. At each decision round $t \in [T]$, the learning agent selects an action $a_t \in [K]$ (i.e., a choice of assets to user $t$) and receives a reward $r_t$ (e.g., the corresponding payoff) that is i.i.d. drawn from an unknown probability distribution associated with the portfolio choice. The goal is to learn to maximize its cumulative payoff.

In practice, applying the celebrated MAB formulation to real-life applications (e.g., the above finance example) needs to deal with both robustness and privacy issues. On the one hand, it is known that financial data is often heavy-tailed (rather than sub-Gaussian) [7, 8]. Moreover, the received payoff data in finance often contains outliers [9] due to data contamination. On the other hand, privacy concern in finance is growing [10–12]. For instance, even if the adversary does not have direct access to the dataset, they are still able to reconstruct other customers' personal information by interacting with the pricing platform and observing its decisions [13].

Motivated by this, a line of work on MABs has focused on designing robust algorithms with respect to heavy-tailed rewards [14], adversary contamination [15, 16], or both [17]. Another line of recent work has studied privacy protection in MABs via different trust models of differential privacy (DP) [18] such as central DP [19, 20], local DP [21, 22] and distributed DP [23, 24]. Moreover, there have also

---

[*]Equal contribution.

37th Conference on Neural Information Processing Systems (NeurIPS 2023).

been recent advances in understanding the close relationship between robustness and privacy for the mean estimation problem (e.g., robustness induces privacy [25, 26] and vice versa [27]). In light of this, a fundamental question we are interested in this paper is:

*Is there a simple algorithm that can tackle privacy and robustness in MABs simultaneously?*

**Our contributions.** We give an affirmative answer to it by showing that a unified truncation-based algorithm could achieve a nearly optimal trade-off between regret, privacy, and robustness for MABs under two different heavy-tailed settings. The key intuition is that reward truncation not only helps to reduce outliers (due to both heavy tails and contamination), but also bound its sensitivity (useful for DP guarantees). To make our intuition rigorous, we take the following principled approaches.

**(i)** We first establish the minimax regret lower bound for private and robust MABs, i.e., heavy-tailed MABs with both privacy constraints and Huber's contamination (see section 4) where the reward distribution of each action has finite $k$-th moment for $k \geq 2$. This characterizes the information-theoretic limit of regret with respect to privacy budget, contamination level and heavy-tailedness.

**(ii)** To match the lower bound, we first propose a meta-algorithm (see section 5), which builds upon the idea of batched successive elimination and relies on a generic private and robust mean estimation sub-routine denoted by PRM. Then, for two different settings of (heavy-tailed) reward distributions (i.e., finite raw or central moments), we propose corresponding schemes for the sub-routine PRM, both of which require truncation and the Laplace mechanism to guarantee robustness and privacy, simultaneously. Armed with these, our meta-algorithm can enjoy nearly matching regret upper bounds (see section 6). Experimental studies also corroborate our theoretical results.

**(iii)** Along the way, several results could be of independent interest. In particular, our proposed PRM shows that reward truncation is sufficient to help achieve the optimal high-probability concentration for private and robust mean estimation in the one-dimension case. Moreover, without contamination, our regret upper bounds not only match the optimal one for private heavy-tailed MABs with finite raw moments, but also provide the first results for the case with finite central moments, hence a complete study for private bandits as well.

Due to space limit, **experiments (section A in Appendix), technical lemmas, and all proofs** are relegated to Appendix.

## 2  Related Work

**Robust MABs.** The studies on robust bandits can be largely categorized into two groups. The first group of work mainly focuses on the setting where the total contamination is bounded, i.e., the cumulative difference between observed reward and true reward is bounded by some constant [15]. The second group considers Huber's $\alpha$-contamination model [28] (which is also the focus of our paper) or a similar $\alpha$-fraction model. In these cases, the reward for each round can be contaminated by an arbitrary distribution with probability $\alpha \in [0, 1/2)$ [29, 16, 30], or at most $\alpha$-fraction of the rewards are arbitrarily contaminated [31]. The existing work in this group has mainly focused on the light-tailed setting where the true inlier distribution is Gaussian or sub-Gaussian and uses a robust median or trimmed-mean estimator. A very recent work [17] studies the (non-private) setting where the inlier distribution only has finite variance and uses Huber's estimator to establish problem-dependent bounds. In contrast, we take the perspective of minimax regret, i.e., problem-independent bounds, and also account for privacy protection.

**Private MABs.** In addition to the above mentioned results on private MABs with light-tailed rewards, [22] studies private heavy-tailed MABs with finite raw moments under both central and local models of DP. However, how to design private algorithms for heavy-tailed distributions with finite *central* moments is left as an important problem. In this paper, we resolve the problem as a byproduct of our main results.

**Robust and private mean estimation.** Our work is also related to robust and private mean estimation, especially the one-dimensional case. On the robustness side with Huber's model, a high-probability concentration bound for the median of Gaussian (hence the mean by symmetry) is first established in [32]. Recently, [30] gives a high probability mean concentration via a trimmed-mean estimator for general sub-Gaussian inlier distributions while [33] focuses on the heavy-tailed setting. We refer the reader to the survey paper [34] on high-dimensional robust statistics focusing on robust mean estimation. On the robustness side with heavy-tail, we refer the reader to the survey paper

[35] for mean estimation and regression under heavy-tailed distributions. Besides median-of-means techniques and trimmed mean we mentioned above to handle heavy-tailed data, Catoni's estimator is a very different estimator for heavy tail [36, 37] and also used in bandits [14]. On the privacy side, one close work is [38], which presents the first high-probability mean concentration for private heavy-tailed distributions with finite central moments (via a medians-of-means approach). It is worth noting that there are recent exciting advances in understanding the close relationship between robustness and privacy (e.g., robustness induces privacy [25, 26] and vice versa [27]). From this aspect, our results imply that for the one-dimensional mean estimation problem, truncation alone suffices to help to achieve both.

**Concurrent and independent work.** While preparing this submission, we have noticed that [39] also studies private and robust bandits problems under the Huber model. However, there are many differences between our work and theirs. First, for the differential privacy and bandit models, we investigate the multi-armed bandit problem under central differential privacy while they study stochastic linear bandits under the local differential privacy model. For the assumption of rewards, we focus on the heavy-tailed cases where the distribution of reward has finite $k$-th raw moment and central moment for $k \geq 2$ while they essentially assume that the rewards are bounded[2].

## 3 Preliminaries

In this section, we first formally introduce our problem of private and robust MABs and subsequently present its corresponding regret notions.

### 3.1 Private and Robust MABs

In this paper, we consider the multi-armed bandits problem where the agent interacts with the environment for $T$ rounds. In each round $t$, the agent chooses an action $a_t \in [K]$ and then standard reward $r_t$ is generated independently from inlier distribution. After contamination, the agent observes contaminated reward $x_t$.

As mentioned before, by robustness, we aim to handle both reward contamination and possible heavy-tailed inlier distributions. To this end, we first introduce the following two classes of heavy-tailed reward distributions.

**Definition 3.1** (Finite $k$-th raw moment). *A distribution over $\mathbb{R}$ is said to have a finite $k$-th raw moment if it is within*

$$\mathcal{P}_k = \left\{ P : \mathbb{E}_{X \sim P} \left[ |X|^k \right] \leq 1 \right\}, \quad k \geq 2, \tag{1}$$

*where $k$ is considered fixed but arbitrary.*

**Definition 3.2** (Finite $k$-th central moment [40, 38]). *A distribution over $\mathbb{R}$ is said to have a finite $k$-th central moment if it is within*

$$\mathcal{P}_k^c = \left\{ P : \mathbb{E}_{X \sim P} \left[ |X - \mu|^k \right] \leq 1 \right\}, \quad k \geq 2, \tag{2}$$

*where $\mu := \mathbb{E}_{X \sim P}[X] \in [-D, D]$ and $D \geq 1$.*

The relationship between the finite raw moment and the finite central moment has been discussed in [22]. We further consider the celebrated Huber contamination model [28] in heavy-tailed MABs.

**Definition 3.3** (Heavy-tailed MABs with Huber contamination). *Given the corruption level $\alpha \in [0, 1/2)$. For each round $t \in [T]$, the observed reward[3] $x_t$ for action $a_t$, is sampled independently from the true distribution $P_{a_t} \in \mathcal{P}_k$ (or $P_{a_t} \in \mathcal{P}_k^c$) with probability $1 - \alpha$; otherwise is sampled from some arbitrary and unknown contamination distribution $G_{a_t} \in \mathcal{G}$.*

In addition to robustness, we also consider the privacy protection in MABs via the lens of DP. In particular, we consider the standard central model of DP for MABs (e.g., [41]), where the learning agent has access to users' raw data (i.e., rewards) and guarantees that its output (i.e., sequence

---

[2]Note that although the general problem formulation in [39] considers sub-Gaussian rewards, their privacy guarantee only holds for bounded rewards.

[3]Here we use $x_t$ in the contaminated case to distinguish with standard reward $r_t$.

of actions) are indistinguishable in probability on two neighboring reward sequences. Due to contamination, the reward data accessed by the learning agent at round $t$ could have already been contaminated. More precisely, we let $\mathcal{D}_T = (x_1, \ldots, x_T) \in \mathbb{R}^T$ be a reward sequence generated in the learning process and $\mathcal{M}(\mathcal{D}_T) = (a_1, \ldots, a_T) \in [K]^T$ to denote the sequence of all actions recommended by a learning algorithm $\mathcal{M}$. With this setup, we have the following formal definition.

**Definition 3.4** (Differential Privacy for MABs). *For any $\epsilon > 0$, a learning algorithm $\mathcal{M} : \mathbb{R}^T \to [K]^T$ is $\epsilon$-DP if for all sequences $\mathcal{D}_T, \mathcal{D}_T' \in \mathbb{R}^T$ differing only in a single element and for all events $E \subset [K]^T$, we have*

$$\mathbb{P}\left[\mathcal{M}(\mathcal{D}_T) \in E\right] \leq e^\epsilon \cdot \mathbb{P}\left[\mathcal{M}\left(\mathcal{D}_T'\right) \in E\right].$$

In other words, we protect the privacy of any individual user who interacts with the learning agent in the sense that an adversary observing the output of the learning agent (i.e., a sequence of actions) cannot infer too much about whether any particular individual has participated in this process, or the specific reward feedback of this individual. In this paper, we will leverage the well-known Laplace mechanism to guarantee differential privacy.

**Definition 3.5** (Laplace Mechanism). *Given a function $f : \mathcal{X}^n \to \mathbb{R}^d$, the Laplacian mechanism is given by*

$$\mathcal{M}_L(\mathcal{D}_n, f, \epsilon) = f(\mathcal{D}_n) + (Y_1, Y_2, \cdots, Y_d),$$

*where $Y_i$ is i.i.d. drawn from a Laplacian Distribution[4] $Lap(\frac{\Delta_1(f)}{\epsilon})$, where $\Delta_1(f)$ is the $\ell_1$-sensitivity of the function $f$, i.e., $\Delta_1(f) = \sup_{\mathcal{D}_n \sim \mathcal{D}_n'} ||f(\mathcal{D}_n) - f(\mathcal{D}_n')||_1$. Then, for any $\epsilon > 0$, Laplacian mechanism satisfies $\epsilon$-DP.*

In the following sections, for brevity, we will simply use *private and robust MABs* to refer to our setting, i.e., heavy-tailed MABs with Huber contamination and privacy constraints.

### 3.2 Regrets for Private and Robust MABs

In the contamination case, the standard regret using observed (contaminated) rewards $\{x_t\}_{t \in [T]}$ is ill-defined [31]. Instead, the literature focuses on the *clean regret*, that is, to compete with the best policy in hindsight as measured by the expected true uncontaminated rewards [31, 17, 42]. Hence, let $\mu_a$ be the mean of the inlier distribution of arm $a \in [K]$ and $\mu^* = \max_{a \in [K]} \mu_a$. We also let $\Pi^\epsilon$ be the set of all $\epsilon$-DP MAB algorithms and $\mathcal{E}_{\alpha,k}$ be the set of all instances of heavy-tailed MABs (with parameter $k$) with Huber contamination (of level $\alpha$).

**Definition 3.6** (Clean Regret). *Fix an algorithm $\pi \in \Pi^\epsilon$ and an instance $\nu \in \mathcal{E}_{\alpha,k}$. Then, the clean regret of $\pi$ under $\nu$ is given by $\mathcal{R}_T(\pi, \nu) := \mathbb{E}_{\pi,\nu}[T\mu^* - \sum_{t=1}^T \mu_{a_t}]$.*

Note that here the expectation is taken over the randomness generated by the *contaminated* environment and $\epsilon$-DP MAB algorithm while the means are of the true inlier distributions.

To capture the intrinsic difficulty of the private and robust MAB problem, we are also interested in its minimax regret.

**Definition 3.7** (Minimax Regret). *The minimax regret of our private and robust MAB problem is defined as $\mathcal{R}_{\epsilon,\alpha,k}^{minimax} := \inf_{\pi \in \Pi^\epsilon} \sup_{\nu \in \mathcal{E}_{\alpha,k}} \mathbb{E}_{\pi,\nu}[T\mu^* - \sum_{t=1}^T \mu_{a_t}]$.*

## 4 Lower Bound

We start with the following lower bound on the minimax regret, which characterizes the fundamental impact of privacy budget (via $\epsilon$), contamination level (via $\alpha$) and heavy-tailedness of rewards (via $k$) in the regret. Note that this lower bound is mainly established under Definition 3.1, which in turn also serves as one valid lower bound for the central moment case in Definition 3.2.

**Theorem 4.1.** *Consider a private and robust MAB problem where inlier distributions have finite $k$-th raw (or central) moments ($k \geq 2$). Then, its minimax regret satisifes*

$$\mathcal{R}_{\epsilon,\alpha,k}^{minimax} = \Omega\left(\sqrt{KT} + (K/\epsilon)^{1-\frac{1}{k}} T^{\frac{1}{k}} + T\alpha^{1-\frac{1}{k}}\right).$$

---

[4]For a parameter $\lambda$, the Laplacian distribution has the density function $\mathrm{Lap}(\lambda)(x) = \frac{1}{2\lambda} \exp(-\frac{|x|}{\lambda})$.

**Algorithm 1** Private and Robust Arm Elimination

---

1: **Input:** Number of arms $K$, time horizon $T$, privacy budget $\epsilon$, Huber parameter $\alpha \in (0, 1/2)$, error probability $\delta \in (0, 1]$, inlier distribution parameters i.e., $k$ and optional $D$.
2: Initialize: $\tau = 0$, active set of arms $\mathcal{S} = \{1, \cdots, K\}$.
3: **for** batch $\tau = 1, 2, \ldots$ **do**
4:     Set batch size $B_\tau = 2^\tau$.
5:     **if** $B_\tau < \mathcal{T}$ **then**
6:         Randomly select an action $a \in [K]$.
7:         Play action $a$ for $B_\tau$ times.
8:     **else**
9:         **for** each active arm $a \in \mathcal{S}$ **do**
10:           **for** $i$ from 1 to $B_\tau$ **do**
11:             Pull arm $a$, observe contaminated reward $x_i^a$.
12:             If total number of pulls reaches $T$, **exit**.
13:           **end for**
14:           Set truncation threshold $M_\tau$.
15:           Set additional parameters $\Phi$.
16:           Compute estimate $\widetilde{\mu}_a = \mathtt{PRM}(\{x_i^a\}_{i=1}^{B_\tau}, M_\tau, \Phi)$.
17:         **end for**
18:         Set confidence radius $\beta_\tau$.
19:         Let $\widetilde{\mu}_{\max} = \max_{a \in \mathcal{S}} \widetilde{\mu}_a$.
20:         Remove all arms $a$ from $\mathcal{S}$ s.t. $\widetilde{\mu}_{\max} - \widetilde{\mu}_a > 2\beta_\tau$.
21:     **end if**
22: **end for**

---

This lower bound basically takes a maximum of three terms. The first term comes from the standard regret for Gaussian rewards, the second one captures the additional cost in regret due to privacy and heavy-tailed rewards, and the last term indicates the additional cost in regret due to contamination and heavy-tailed rewards. Note that, for a given $k$, the impact of privacy and contamination is separable. It would also be helpful to compare our lower bound with the related ones, as discussed below.

**Remark 4.2.** *First, when $k = \infty$ and $\alpha = 0$, our lower bound recovers the state-of-the-art lower bound for private MABs with sub-Gaussian rewards [20]; Second, when there is no privacy protection, a very recent work [17] establishes a* problem-dependent *regret lower bound for robust MABs while we are interested in the problem-independent lower bound with further privacy protection.*

## 5 Our Approach: A Meta-Algorithm

In this section, we first introduce a meta-algorithm for private and robust MABs, which not only allows us to tackle inlier distributions with bounded raw or central moments in a unified way, but also highlights the key component, i.e., a private and robust mean estimation sub-routine building on the main idea of reward truncation.

Our meta-algorithm, at a high level, can be viewed as a batched version of the celebrated successive arm elimination [43] along with a private and robust mean estimation sub-routine PRM (see Algorithm 1). That is, it divides the time horizon $T$ into batches with exponentially increasing size and eliminates sub-optimal arms successively based on the mean estimate via PRM. More specifically, based on the batch size, it consists of two phases. That is, when the batch size is less than a threshold $\mathcal{T}$, it simply recommends actions randomly (line 5-7) (more on this will be explained soon). Otherwise, for each active arm $a$ in batch $\tau$, it first prescribes $a$ to a batch of $B_\tau = 2^\tau$ fresh *new* users (i.e., "doubling") and observes possibly contaminated rewards (line 8). Then, it calls the sub-routine PRM to compute a private and robust mean estimate for each active arm $a$ (line 12). In particular, it *only* uses the rewards within the most recent batch (i.e., "forgetting") along with a proper reward truncation threshold $M_\tau$. Finally, it adopts the classic idea of arm elimination with a proper choice of confidence radius $\beta_\tau$ to remove sub-optimal arms with high confidence (line 18-20).

We now provide more intuitions behind our algorithm design by highlighting how its main components work in concert. *First,* the reason behind the first phase (i.e., $B_\tau \leq \mathcal{T}$), named forced exploration, is necessary as it ensures that concentration is satisfied so that the optimal arm will not be eliminated.

---

**Algorithm 2** PRM for the finite raw moment case

---
1: **Input:** A collection of data $\{x_i\}_{i=1}^n$, truncation parameter $M$, additional parameters $\Phi = \{\epsilon\}$.
2: **for** $i = 1, 2, \ldots, n$ **do**
3:    Truncate data $\bar{x}_i = x_i \cdot \mathbb{1}_{\{|x_i| \le M\}}$.
4: **end for**
5: Return private estimate $\widetilde{\mu} = \frac{\sum_{i=1}^n \bar{x}_i}{n} + \mathrm{Lap}(\frac{2M}{n\epsilon})$.

---

This is due to the fact that in the contamination case, a well-behaved concentration only kicks in when the number of samples is larger than a threshold. Thus, one cannot adopt arm elimination in this phase since it might eliminate the optimal arm. Note that, instead of our choice of random selection, one can also use other methods for the first phase (see Remark 5.1 below). *Second,* for the second phase, the idea of doubling batching and forgetting is the key to achieving privacy with a minimal amount of noise (hence better regret). This is because now any single reward feedback only impacts one computation of estimate. This is in sharp contrast to standard arm elimination (e.g., [43]) where each mean estimate is based on all samples so far (as no batching is used), and hence a single reward change could impact $O(T)$ mean estimations[5]. *Third,* the simple idea of reward truncation in PRM turns out to be extremely useful for both robustness and privacy. On the one hand, truncation helps to reduce the impact of outliers (due to both heavy tails and contamination); On the other hand, truncation also helps to bound the sensitivity, which is necessary for privacy. In fact, as we will show later, a well-tuned truncation threshold enables us to achieve a near-optimal trade-off between regret, privacy and robustness. *Finally,* in contrast to the first phase, we can now eliminate sub-optimal arms with high confidence due to the high probability concentration of mean estimate when batch size is larger than $\mathcal{T}$ (more details will be given later for specific choices of PRM and hence the choice of $\mathcal{T}$).

We choose to use the successive elimination (SE) technique here because it suffices to enable us to achieve optimal order-wise regret later. In fact, once armed with our novel PRM module, one can also use other exploration strategies like UCB in [20]. One difference is that now instead of first pulling each arm once, it needs to pull each arm $\mathcal{T}$ times to ensure that concentration kicks in later. This is in fact not surprising since on the high-level, the analysis of SE and UCB is very similar, i.e., doubling (batching) and forgetting. We also note that instead of UCB, one can also adapt it to the Thompson sampling strategy, e.g., [45], with our PRM module. Again, the key idea is batching and forgetting.

**Remark 5.1.** *The algorithmic choice of the first phase can be flexible. For example, instead of playing a randomly selected action for the whole batch, one can choose to play a randomly selected action for each round. Moreover, one can also choose to be greedy or probabilistically greedy with respect to the mean estimate by* PRM, *which also only uses the rewards collected within the last batch for each arm. All of these choices have the same theoretical guarantees, though some will help to improve the empirical performance.*

We then present the following remark that places our meta-algorithm in the existing literature.

**Remark 5.2** (Comparison with existing literature). *For private MABs (without contamination), the state-of-the-art also builds upon the idea of batching and forgetting [19, 20, 24] to achieve optimal regret. For robust MABs (without privacy), existing works take different robust mean estimations. For example, both [31, 30] use a trimmed mean estimator for sub-Gaussian inlier distributions while [17] adopts Huber's estimator to handle inlier distributions with only bounded variance. We are the first to study privacy and robustness simultaneously, via a simple truncation-based estimator, which in turn reveals the close relationship between privacy and robustness in MABs. This complements the recent advances in capturing the connection between these two in (high-dimensional) statistics [25, 27].*

## 6   Upper Bounds

In this section, we establish the regret upper bounds for two specific instantiations of our meta-algorithm – one for the finite raw moment case and another for the finite central moment case. The results could match our lower bound up to a log factor on $T$, demonstrating their near-optimality.

---

[5]One can use the tree-based mechanism [44] to reduce it to $O(\log T)$, but it is still sub-optimal [19].

## 6.1 Finite Raw Moment Case

In this section, we will focus on private and robust MABs where the inlier distributions have a finite $k$-th raw moment as given by Definition 3.1. In particular, we first introduce the choice of PRM in this case (see Algorithm 2) and establish its concentration property, which plays a key role in our implementation of meta-algorithm.

The PRM in Algorithm 2 is simply a truncation-based Laplace mechanism. That is, it first truncates all the received data with the threshold $M$ (line 3). Then, Laplace noise is added to the empirical mean to preserve privacy (line 5). We highlight again that truncation here helps with both robustness (via removing outliers) and privacy (via bounding the sensitivity of empirical mean).

As in the standard algorithm design of MABs, the key is to utilize the concentration of the mean estimator. To this end, we first give the following high-probability concentration result for the mean estimate returned by PRM in Algorithm 2.

**Theorem 6.1** (Concentration of Mean Estimate). *Given a collection of Huber-contaminated data $\{x_i\}_{i=1}^n$ where the inlier distribution satisfies Definition 3.1 with mean $\mu$, let $\widetilde{\mu}$ be the mean estimate by Algorithm 2. Then, for any privacy budget $\epsilon > 0$ and $\delta \in (0,1)$, the following results hold:*

***Uncontaminated case.*** *For $\alpha = 0$, we have $|\widetilde{\mu} - \mu| = O\left(\sqrt{\frac{\log(1/\delta)}{n}} + \frac{M\log(1/\delta)}{n\epsilon} + \frac{1}{M^{k-1}}\right)$, with probability at least $1 - \delta$. Thus, choosing the truncation threshold $M = \Theta\left(\frac{n\epsilon}{\log(1/\delta)}\right)^{\frac{1}{k}}$ yields $|\widetilde{\mu} - \mu| = O\left(\sqrt{\frac{\log(1/\delta)}{n}} + \left(\frac{\log(1/\delta)}{n\epsilon}\right)^{1-\frac{1}{k}}\right)$.*

***Contaminated case.*** *For $0 < \alpha \leq \alpha_1 \in (0, 1/2)$ and $n = \Omega\left(\frac{\log(1/\delta)}{\alpha_1}\right)$, we have the following with probability at least $1 - \delta$, $|\widetilde{\mu} - \mu| = O\left(\sqrt{\frac{\log(1/\delta)}{n}} + \frac{M\log(1/\delta)}{n\epsilon} + \frac{1}{M^{k-1}} + \alpha_1 M\right)$. Therefore, choosing the truncation threshold $M = \Theta\left(\min\left\{\left(\frac{n\epsilon}{\log(1/\delta)}\right)^{\frac{1}{k}}, \alpha_1^{-\frac{1}{k}}\right\}\right)$, yields $|\widetilde{\mu} - \mu| \leq \beta$, where $\beta = O\left(\sqrt{\frac{\log(1/\delta)}{n}} + \left(\frac{\log(1/\delta)}{n\epsilon}\right)^{1-\frac{1}{k}} + \alpha_1^{1-\frac{1}{k}}\right)$.*

With the above result, several remarks are in order. *First,* for the uncontaminated case, our concentration result consists of the standard sub-Gaussian term and a new one due to privacy and heavy-tailed data. It can be translated into a sample complexity bound, i.e., to guarantee $|\widetilde{\mu} - \mu| \leq \eta$ for any $\eta \in (0,1)$, it requires the sample size to be $n \geq O(\frac{\log(1/\delta)}{\eta^2} + \frac{\log(1/\delta)}{\epsilon\eta^{k/(k-1)}})$, which is optimal since it matches the lower bound for private heavy-tail mean estimation (cf. Theorem 7.2 in [46]). *Second,* for the contaminated case, it has an additional bias term $O(\alpha_1^{1-1/k})$, which is also known to be information-theoretically optimal [47]. Thus, via truncation, the PRM given by Algorithm 2 achieves the *optimal* trade-off between accuracy, privacy and robustness, which in turn shows its potential to be integrated into our meta-algorithm. Note that $\alpha_1$ here is any upper bound (e.g., estimate) on the true contamination level $\alpha$.

Now, based on the concentration result, we can set other missing parameters in our meta-algorithm accordingly. In particular, we have the following theorem that states the specific instantiation along with its performance guarantees.

**Theorem 6.2** (Performance Guarantees). *Consider a private and robust MAB with inlier distributions satisfying Definition 3.1 and $0 < \alpha \leq \alpha_1 \in (0, 1/2)$. Let Algorithm 1 be instantiated with Algorithm 2 and $M_\tau$, $\beta_\tau$ be given by Theorem 6.1 with $n$ replaced by $B_\tau$. Set $\mathcal{T} = \Omega(\frac{\log(1/\delta)}{\alpha_1})$ and $\delta = 1/T$. Then Algorithm 1 is $\epsilon$-DP with its regret upper bound*

$$\mathcal{R}_T = O\left(\sqrt{KT\log T} + \left(\frac{K\log T}{\epsilon}\right)^{\frac{k-1}{k}} T^{\frac{1}{k}} + T\alpha_1^{1-\frac{1}{k}} + \frac{K\log T}{\alpha_1}\right).$$

The above theorem presents the first achievable regret guarantee for private and robust bandits. The first three terms match our lower bound in Theorem 4.1 up to $\log T$ factor. The last additive term is

---

**Algorithm 3** PRM for the finite central moment case

---

1: **Input:** A collection of data $\{x_i\}_{i=1}^{2n}$, truncation parameter $M$, additional parameters $\Phi = \{\epsilon, D, r\}, r \in \mathbb{R}$.
2: // First step: initial estimate
3: $B_j = [j, j + r), j \in \mathcal{J} = \{-D, -D + r, \ldots, D - r\}$.
4: Compute private histogram using the first fold of data: $\widetilde{p}_j = \frac{\sum_{i=1}^n \mathbb{1}_{\{X_i \in B_j\}}}{n} + \text{Lap}\left(\frac{2}{n\epsilon}\right)$.
5: Get the initial estimate $J = \arg\max_{j \in \mathcal{J}} \widetilde{p}_j$.
6: // Second step: final estimate
7: Get final estimator using the second fold of data: $\widetilde{\mu} = J + \frac{1}{n}\sum_{i=n+1}^{2n}(X_i - J)\mathbb{1}_{\{|X_i - J| \le M\}} + \text{Lap}\left(\frac{2M}{n\epsilon}\right)$.

---

mainly due to the fact that the mean concentration result only holds when the sample size is larger than $\mathcal{T} = \Omega(\frac{\log(1/\delta)}{\alpha_1})$. As a result, each sub-optimal has to be played at least $\Omega(\frac{\log(1/\delta)}{\alpha_1})$ times. However, for a sufficiently large $T$ and a constant $\alpha$, the last term is dominated by other terms. The last term is also exactly the reason that we choose an upper bound $\alpha_1$ on the actual contamination level $\alpha$ and state the upper bound results in terms of $\alpha_1$ rather than $\alpha$. That is, for a very small but non-zero $\alpha$, one can choose a larger $\alpha_1$ to balance the regret. This subtle issue is also mentioned in one nice related work [42], see the remark after Theorem 7.4 and Remark 5.4 in their work.

**Remark 6.3.** *For the uncontaminated case with $\alpha = 0$, using the uncontaminated concentration bound in Theorem 6.1 and the same analysis, we achieve a regret upper bound $O(\sqrt{KT\log T} + (\frac{K\log T}{\epsilon})^{\frac{k-1}{k}}T^{\frac{1}{k}})$, which also matches the lower bound in Theorem 4.1 up to a factor of $O(\log T)$.*

## 6.2 Finite Central Moment Case

The setting in the last section for the finite raw moment case may not be entirely satisfactory as it essentially assumes that the mean of arms is bounded within a small range (hence the sub-optimal gaps). Thus, in this section, we turn to private and robust MABs where the inlier distributions have a finite $k$-th central moment as given by Definition 3.2 for a reward distribution with large mean but small variability around the mean. To this end, we first need a new PRM, since now simply truncating around zero as in Algorithm 2 will not work well (as discussed in detail in Remark 6.13).

Our new PRM is presented in Algorithm 3, which consists of two steps. The intuition is simple: the first step aims to have a rough estimate of the mean, which is necessary since now the mean could be far away from zero. Then, in the second step, it truncates around the initial estimate to return the final result. More specifically, in the first step, we first construct bins over the range $[-D, D]$, which is assumed to contain the true mean by Definition 3.2. Then, we compute the private histogram via the Laplace mechanism. The initial estimate is given by the left endpoint of the bin that has the largest empirical mass. Next, in the second step, it simply truncates around the initial estimate and again adds Laplace noise for privacy.

**Remark 6.4.** *It is worth noting that a similar idea of two-step estimation has been used in previous work on robust mean estimation in the one-dimensional heavy-tailed case [33, 38, 40]. However, there are several differences in our algorithm design and analysis. In particular, while [33] considers mean estimation under Huber's model without privacy constraints, we further impose differential privacy requirements. As a result, the estimates for both two steps are in different forms in our case compared to [33], though they share the same high-level intuition. On the other hand, while [38] considers mean estimation under differential privacy, there is no consideration of Huber contamination as in our case. Moreover, our second estimate is based on truncation while their method is via medians-of-means. In fact, as will be shown later (see Remark 6.7), when our result reduces to the uncontaminated case, it achieves improvement over the one in [38]. Finally, [40] considers both Huber contamination and* local *differential privacy, and establishes the corresponding mean square error (MSE). In contrast, we consider the* central *differential privacy and aim to establish a* high-probability tail concentration. *To this end, we take a different truncation method (i.e., using an indicator function in Line 7) compared to the one in [40] (Section 3.2 in its first arxiv version).*

As before, we first present the concentration property of our new PRM, which will manifest in the specific instantiation of our meta-algorithm. In particular, we first give the following general theorem and then state two more detailed corollaries.

**Theorem 6.5** (Concentration of Mean Estimate). *Given a collection of Huber-contaminated data $\{x_i\}_{i=1}^{2n}$ where the inlier distribution satisfies Definition 3.2 with mean $\mu$, let $\widetilde{\mu}$ be the mean estimate by Algorithm 3. For any $\alpha \leq \alpha_1 \in (0, \alpha_{\max})$, $\epsilon \in (0, 1]$ and $\delta \in (0, 1)$, there exist some constants $\mathcal{T}(\alpha_1, \epsilon, \delta)$, $r$, $M$ and $D \geq 2r$ such that for all $n \geq \mathcal{T}(\alpha_1, \epsilon, \delta)$, with probability at least $1 - \delta$*

$$|\widetilde{\mu} - \mu| \leq O\left( \sqrt{\frac{\log(1/\delta)}{n}} + \frac{M \log(1/\delta)}{n\epsilon} + \frac{1}{M^{k-1}} + \alpha_1 M \right),$$

*where $\alpha_{\max} < 1/2$ is the breakdown point.*

The above theorem follows the same pattern as the one for the raw moment case (Theorem 6.1). The key differences are the threshold value $\mathcal{T}(\alpha, \epsilon, \delta)$ and the breakdown point $\alpha_{\max}$, which are summarized in the following results.

**Corollary 6.6** (Mean Concentration, $\alpha = 0$). *Let the same assumptions in Theorem 6.5 hold. For any $\epsilon \in (0, 1]$, setting $r = 10^{1/k}$ and $M = \Theta\left( \frac{n\epsilon}{\log(1/\delta)} \right)^{1/k}$, then for all $n \geq \Omega\left( \log(D/\delta)/\epsilon \right)$ and $D \geq 2r$, we have that for any $\delta \in (0, 1)$, with probability at least $1 - \delta$, $|\tilde{\mu} - \mu| \leq \beta$ where $\beta = O\left( \sqrt{\frac{\log(1/\delta)}{n}} + \left( \frac{\log(1/\delta)}{n\epsilon} \right)^{1-\frac{1}{k}} \right)$. In other words, taking number of samples $n$ such that $n \geq O\left( \frac{\log(1/\delta)}{\eta^2} + \frac{\log(1/\delta)}{\epsilon\eta^{\frac{k}{k-1}}} + \frac{\log(D/\delta)}{\epsilon} \right)$ we have $|\tilde{\mu} - \mu| \leq \eta$ with probability at least $1 - \delta$.*

**Remark 6.7.** *The above lemma strictly improves the result in [38, Theorem 3.5][6]. In particular, it uses the method of medians-of-means and achieves $\frac{\log D \cdot \log(1/\delta)}{\epsilon}$ for the third term. In contrast, our third term is additive rather than multiplicative. In fact, our concentration is optimal, which matches the lower bound for the one-dimensional case (see [46, Theorem 7.2]).*

**Corollary 6.8** (Mean Concentration, $\alpha > 0$). *Let the same assumptions in Theorem 6.5 hold. For any $\epsilon \in (0, 1]$ and $0 < \alpha \leq \alpha_1 \in (0, 0.133)$, we let $r = \iota^{1/k}$ where $\iota = \frac{1-\alpha}{0.249-\alpha}$ and $M = \Theta(\min\{\left( \frac{n\epsilon}{\log(1/\delta)} \right)^{1/k}, (\alpha_1)^{-1/k}\})$. Then, there exists constant $c_1$, for all $n$ such that $n \geq \mathcal{T} = \Omega(\max\{ \frac{\iota \log(1/\delta)}{\epsilon}, \frac{c_1 \log(D/\delta)}{\epsilon}, \frac{\log(1/\delta)}{\alpha_1^2} \})$ and $D \geq 2r$, we have that for any $\delta \in (0, 1)$, with probability at least $1 - \delta$, $|\tilde{\mu} - \mu| \leq \beta$ with $\beta = O\left( \sqrt{\frac{\log(1/\delta)}{n}} + \left( \frac{\log(1/\delta)}{n\epsilon} \right)^{1-\frac{1}{k}} + \alpha_1^{1-\frac{1}{k}} \right)$.*

**Remark 6.9.** *The above concentration has the same form as the one in Theorem 6.1. Specifically, for a large sample size $n$, it has the optimal concentration (for small $\alpha$). The threshold $\mathcal{T}$ on $n$ depends on both $\alpha, \epsilon$ now. We note that even for the sub-Gaussian inlier distributions without privacy protection, the existing concentration also has a threshold $\mathcal{T} = \frac{\log(1/\delta)}{\alpha_1^2}$ (see Lemma 4.1 in [30]). The finite central moment assumption will help us get logarithmic results for sample complexity on $D$ of $n \geq \Omega\left( \log(D/\delta)/\epsilon \right)$ which is better than the polynomial results on $D$ from the finite raw moment. Actually, finding a private estimator that achieves an error on the logarithmic of the range of the parameter is significant in the topic of DP estimation (see [48–50] on the importance of the problem).*

Now, we are left to leverage the above two concentration results to design specific instantiations of our meta-algorithm and establish their performance guarantees. Our first instantiation is for the uncontaminated case, i.e., $\alpha = 0$. Therefore, robustness is then only with respect to heavy-tailed rewards while privacy is still preserved.

**Theorem 6.10** (Performance Guarantees, $\alpha = 0$). *Consider a private and robust MAB with inlier distributions satisfying Definition 3.2 and $\alpha = 0$. Let Algorithm 1 be instantiated with Algorithm 3, and $r$, $M_\tau$, $\beta_\tau$ be given by Corollary 6.6 with $n$ replaced by $B_\tau$. Set $\mathcal{T} = \Omega(\frac{\log(D/\delta)}{\epsilon})$ and $\delta = 1/T$. Then, Algorithm 1 is $\epsilon$-DP with its regret upper bound*

$$\mathcal{R}_T = O\left( \sqrt{KT \log T} + (K \log T/\epsilon)^{\frac{k-1}{k}} T^{\frac{1}{k}} + \gamma \right),$$

*where $\gamma := O\left( KD \log(DT)/\epsilon \right)$.*

---
[6] We also note that the main focus of [38] is not on achieving the optimal estimate though.

**Remark 6.11.** *To the best of our knowledge, this is the first result on private and heavy-tailed bandits with the finite central moment assumption. The state-of-the-art result only focuses on the simpler case, i.e., the finite raw moment assumption [22].*

Finally, armed with Corollary 6.8, we have the second instantiation of our meta-algorithm that deals with the contaminated case.

**Theorem 6.12** (Performance Guarantees, $\alpha > 0$). *Consider a private and robust MAB with inlier distributions satisfying Definition 3.2 and $\alpha \le \alpha_1 \in (0, 0.133)$. Let Algorithm 1 be instantiated with Algorithm 3, and $r$, $\mathcal{T}$, $M_\tau$, $\beta_\tau$ be given by Corollary 6.8 with $n$ replaced by $B_\tau$. Set $\delta = 1/T$, then Algorithm 1 is $\epsilon$-DP with its regret upper bound*

$$\mathcal{R}_T = O\left(\sqrt{KT\log T} + (K\log T/\epsilon)^{\frac{k-1}{k}} T^{\frac{1}{k}} + T\alpha_1^{1-\frac{1}{k}} + \hat{\gamma}\right),$$

*where $\hat{\gamma} := O\left(\frac{DK\log T}{\alpha_1^2} + \frac{\iota DK\log T}{\epsilon} + \frac{DK\log(DT)}{\epsilon}\right)$ and $\iota = \frac{1-\alpha}{0.249-\alpha}$.*

**Remark 6.13.** *The above upper bound also matches our lower bound up to log factor and $O(\hat{\gamma})$, which is dominated by other terms for a sufficiently large $T$ and small $\alpha, D$. A comparison between the results in Theorem 6.2 (Remark 6.3) and Theorem 6.12 (Theorem 6.10) for contaminated case (uncontaminated case), respectively, allows us to address a fundamental question: why we do not first transform the central moment condition to the raw moment condition, and then simply apply Algorithms 1 and 2 to get a regret upper bound? This is because the central moment condition implies $\mathbb{E}[|X|^k] = O(D^k)$. By employing a simple scaling technique, Algorithms 1 and 2 would yield a regret bound of $O(D \cdot Reg)$, where Reg corresponds to the bound stated in Theorem 6.2. Thus, a significant contrast arises when comparing this result with Theorem 6.12, where the dependence on $D$ is only an additive linear term rather than a multiplicative factor of $O(D)$. Consequently, the bounds presented in Theorem 6.12 and Theorem 6.10 prove to be substantially tighter when $D$ assumes larger values, hence highlighting the contributions of our new PRM module in Algorithm 3.*

## 7    Simulations and Conclusion

We refer readers to Appendix A for our simulation results. In this paper, we investigated private and robust multi-armed bandits with heavy-tailed rewards under Huber's contamination model as well as differential privacy constraints. We proposed a meta-algorithm that builds on a private and robust mean estimation sub-routine PRM for different heavy-tailed assumptions of rewards, i.e. finite $k$-th raw moment and finite $k$-th central moment with $k \ge 2$. Moreover, we also established regret upper bounds for these algorithms, which nearly match our derived minimax lower bound.

There remain many open questions in this direction. First, the problem-dependent lower bound is unclear for private and robust bandits but some work has been done for corrupted heavy-tailed bandits (see Theorem 1 in [17]). One interesting future direction is to study how to leverage both the insights in [17] and the lower bound under privacy to derive a problem-dependent lower bound for private and robust bandits. Second, how to design algorithms and analyze the theoretical results for private and robust bandits under other corrupted models such as "bandits with total corruption budget" in [15]. Third, throughout the paper $T$ is known in advance, and it remains to see how to handle the case where $T$ is unknown and infinity.

## 8    Acknowledgments

We thank Mengchu Li for the insightful discussions and for pointing out the first arxiv version of [40]. And we also thank the anonymous NeurIPS reviewers for their feedback. Xingyu Zhou is supported in part by NSF grant CNS-2153220. Di Wang and Yulian Wu were supported in part by the baseline funding BAS/1/1689-01-01, funding from the CRG grand URF/1/4663-01-01, FCC/1/1976-49-01 from CBRC of King Abdullah University of Science and Technology (KAUST). Di Wang was also supported by the funding of the SDAIA-KAUST Center of Excellence in Data Science and Artificial Intelligence (SDAIA-KAUST AI)

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
