# A Experiments

In this section, we empirically evaluate the practical performance of our private and robust arm elimination algorithms, denoted as PRAE-R and PRAE-C. These algorithms employ PRM as a subroutine, which can be either Algorithm 2 for the finite raw moment case or Algorithm 3 for the finite central moment case. We benchmark our algorithms against DPRSE [22], which attains the optimal regret bound for DP heavy-tailed MAB, and RUCB [16], which is a non-private robust algorithm.

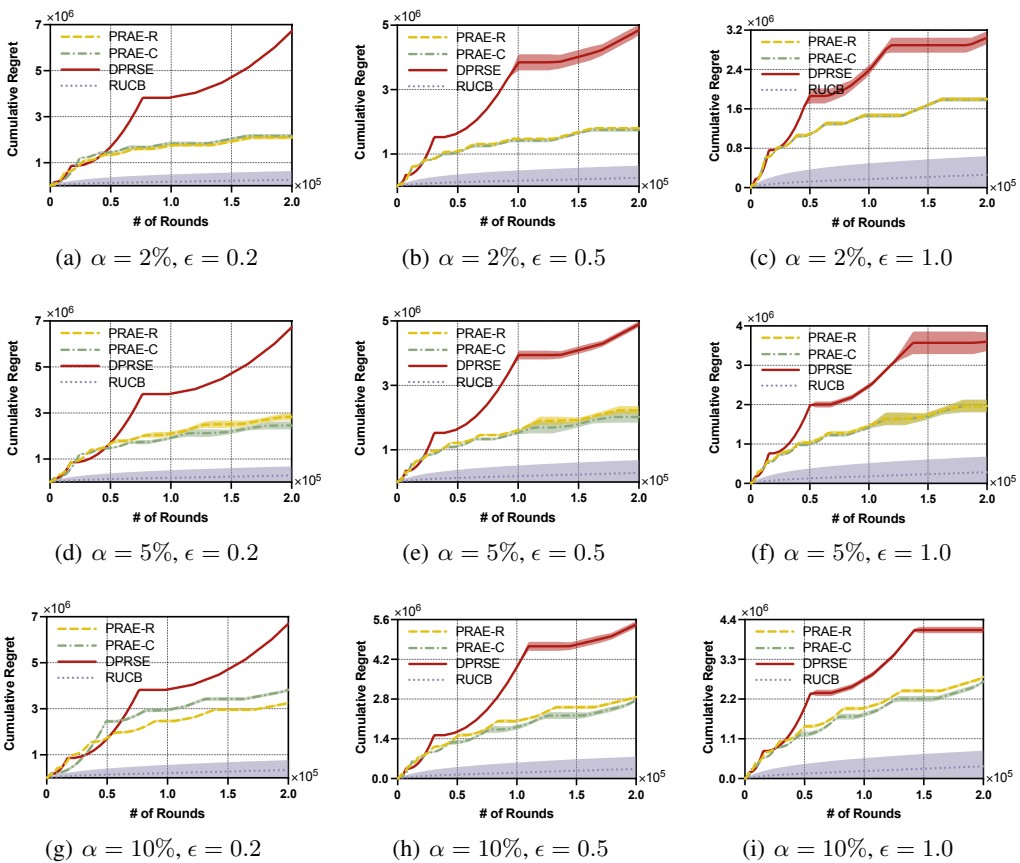

Figure 1: Comparison of cumulative regret for PRAE-R, PRAE-C and DPRSE under Pareto reward.

## A.1 Experiment Setup

We consider the case where there are $K = 11$ arms, and the mean of each arm is within the range of $[0, 100]$. We set the means of the optimal arm and the worst arm to be $100$ and $0$, respectively, and the means of other arms scale linearly with a gap of $10$. Specifically, for each arm $a \in [k]$, we have $\mu_a = 100 - \frac{100(a-1)}{K-1}$. We consider the following two types of heavy-tailed distributions for the true inlier reward generation:

- *Pareto distribution:* For each pull of arm $a$, we generate a reward that is sampled from the distribution $\mu_a + \eta - 2.5$, where $\eta \sim \frac{s x_m^s}{x^{s+1}} \mathbb{1}_{\{x \geq x_m\}}$ for $x \in \mathbb{R}$ and we set the shape parameter $s = 3$ and the scale parameter $x_m = 40$.

- *Student's t-distribution:* For each pull of arm $a$ we generate a reward that is sampled from the distribution $\mu_a + \eta$, where $\eta \sim \frac{\Gamma(\frac{\nu+1}{2})}{\sqrt{\nu \pi} \Gamma(\frac{\nu}{2})} \left(1 + \frac{x^2}{\nu}\right)^{-\frac{\nu+1}{2}}$. Here we set the degree of freedom $\nu = 2.0017$.

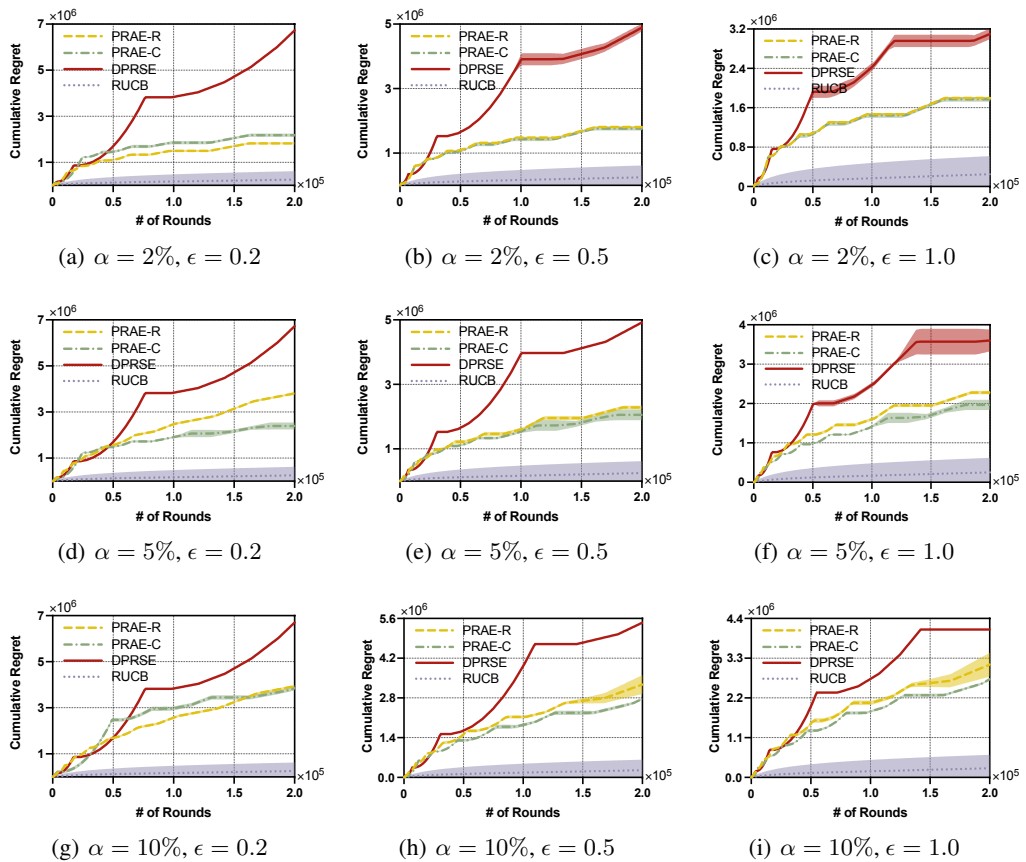

Figure 2: Comparison of cumulative regret for PRAE-R, PRAE-C and DPRSE under Student's $t$ reward.

For both cases, the stochastic rewards have finite second central moment of 35. The main difference between the above two types of distribution is that the Student's $t$-distribution is symmetric while the Pareto distribution is one-sided.

To generate contaminated rewards, we still use Gaussian distribution. For the optimal arm, we set the mean of the Gaussian corruption distribution to be 0. For the other sub-optimal arms, we let the mean of their corrupted reward be 100. This way, the optimal arm is under-evaluated and the non-optimal arms are over-evaluated.

In our experiments, we will vary $\alpha$ and $\epsilon$ in $\{2\%, 5\%, 10\%\}$ and $\{0.2, 0.5, 1\}$, respectively. For each case, we repeat 30 times and set the total number of round $T = 10^5$ (thus we set $\delta = 10^{-5}$). We will report the average of cumulative regrets $\mathcal{R}_T$ with respect to the number of rounds.

### A.2 Results and Discussions

We present the experimental results under Pareto reward and Student's t reward in Figure 1 and Figure 2, respectively. From these results, we can make the following observations:

- Firstly, by comparing the cumulative regret, we see that for all cases, PRAE-R and PRAE-C achieve smaller cumulative regret and thus better expected performance than DPRSE. In particular, when the Huber parameter $\alpha$ increases, DPRSE diverges to a larger regret, while PRAE-R and PRAE-C are only slightly affected. This is because DPRSE adopts more aggressive truncation thresholds that incorporate more outliers. In contrast, the truncation thresholds in PRAE-R and PRAE-C are carefully designed and thus provide robustness against contaminated rewards.

- Secondly, by looking at the error bars, we notice that PRAE-R and PRAE-C have smaller variance compared with both the private baseline and robust baseline under both symmetric and one-sided types of heavy-tailed distribution. In particular, although RUCB provides lower bounds on the performance with privacy for all the settings, PRAE-R and PRAE-C are more stable. The main reason could be that their robust mean estimator is based on median, which incurs smaller estimation error than the estimators developed in our paper.

- Thirdly, for both PRAE-R and PRAE-C, we also observe that when $\epsilon$ is smaller or $\alpha$ is larger, the regret will increase for both types of distributions, which is consistent with the fact that the regret bound is proportional to $1/\epsilon$ and $\alpha$ when $T$ is large enough. Moreover, compared with PRAE-R, we can see that the regret of PRAE-C is lower in most cases. This is due to the fact that the PRM subroutine for PRAE-C leverages the prior information (i.e., range D) of mean for each arm, which could provide tighter performance bound. In summary, all these results corroborate our theoretical analysis.

# B  Useful Lemmas

**Lemma B.1** (Post-Processing [51]). *Let $\mathcal{M} : \mathcal{X} \to \mathcal{Y}$ be a randomized algorithm that is $(\epsilon, \delta)$-differentially private. Let $f : \mathcal{Y} \to \mathcal{Z}$ be an arbitrary randomize mapping. Then $f \circ \mathcal{M} : \mathcal{X} \to \mathcal{Z}$ is $(\epsilon, \delta)$-differentially private.*

**Lemma B.2** (Parallel Composition [52]). *Suppose there are $n$ $\epsilon$-differentially private mechanisms $\{\mathcal{M}_i\}_{i=1}^n$ and $n$ disjoint datasets denoted by $\{D_i\}_{i=1}^n$. Then for the algorithm which applies each $\mathcal{M}_i$ on the corresponding $D_i$, it is $\epsilon$-DP.*

**Lemma B.3** (Markov's inequality). *If $Y \in \mathbb{R}$ is a random variable and $a > 0$, we have*

$$\mathbb{P}\left(|Y| \geq a\right) \leq \frac{\mathbb{E}\left(|Y|^k\right)}{a^k}$$

**Lemma B.4** (Chebyshev's inequality). *For a real-valued random variable $Y \in \mathbb{R}$, $a > 0$ and $k \in \mathbb{N}$, we have*

$$\mathbb{P}(|Y - \mathbb{E}Y| \geq a) = \mathbb{P}\left(|Y - \mathbb{E}Y|^k \geq a^k\right) \leq \frac{\mathbb{E}\left(|Y - \mathbb{E}Y|^k\right)}{a^k}$$

**Lemma B.5** (Tail Bound of Laplacian Vairable [18]). *If $X \sim \text{Lap}(b)$, then*

$$\mathbb{P}(|X| \geq t \cdot b) = \exp(-t).$$

**Lemma B.6** (Hoeffding's inequality). *Let $Z_1, \ldots, Z_n$ be independent bounded random variables with $Z_i \in [a, b]$ for all $i$, where $-\infty < a < b < \infty$. Then*

$$\mathbb{P}\left(\left|\frac{1}{n}\sum_{i=1}^n (Z_i - \mathbb{E}[Z_i])\right| \geq t\right) \leq 2\exp\left(-\frac{2nt^2}{(b-a)^2}\right)$$

**Lemma B.7** (Hölder's Inequality). *Let $X, Y$ be random variables over $\mathbb{R}$, and let $k > 1$. Then,*

$$\mathbb{E}[|XY|] \leq \left(\mathbb{E}\left[|X|^k\right]\right)^{\frac{1}{k}} \left(\mathbb{E}\left[|Y|^{\frac{k}{k-1}}\right]\right)^{\frac{k-1}{k}}$$

**Lemma B.8** (Bernstein's Inequality [53]). *Let $X_1, \cdots X_n$ be $n$ independent zero-mean random variables. Suppose $|X_i| \leq M$ and $\mathbb{E}[X_i^2] \leq s$ for all $i$. Then for any $t > 0$, we have*

$$\mathbb{P}\left\{\left|\frac{1}{n}\sum_{i=1}^n X_i\right| \geq t\right\} \leq 2\exp\left(-\frac{\frac{1}{2}t^2 n}{s + \frac{1}{3}Mt}\right)$$

# C  Proofs of Section 4

**Lemma C.1** (Upper Bound on KL-divergence for Bandits with $\epsilon$-DP [20]). *If $\pi$ is a mechanism satisfying $\epsilon$-DP, then for two instances $\nu_1 = (r_a : a \in [K])$ and $\nu_2 = (r'_a : a \in [K])$ we have*

$$KL\left(\mathbb{P}_{\pi,\nu_1}^T \| \mathbb{P}_{\pi,\nu_2}^T\right) \leq 6\epsilon \mathbb{E}_{\pi,\nu_1}\left[\sum_{t=1}^T TV(r_{a_t} \| r'_{a_t})\right]$$

*where $TV(r_a \| r'_a)$ is the total-variation distance between $r_a$ and $r'_a$.*

**Lemma C.2** (Theorem 5.1 in [54]). *Let $R_1$ and $R_2$ be two distributions on $\mathcal{X}$. If for some $\alpha \in [0, 1/2)$, we have that $TV(R_1, R_1) = \frac{\alpha}{1-\alpha}$, then there exists two distributions on the same probability space $G_1$ and $G_2$ such that*

$$(1 - \alpha)R_1 + \alpha G_1 = (1 - \alpha)R_2 + \alpha G_2.$$

**Proof of Theorem 4.1.** Let $\Pi$ be the set of all policies and $\Pi^\epsilon$ be the set of all $\epsilon$-DP policies. We denote the environment corresponding to the set of $K$-Gaussian reward distributions with means $\mu \in \mathbb{R}^K$ the same variance $\sigma_k^2$ where the value of $\sigma_k$ is determined by $k$ to make the $k$-th raw moments of the distributions are bounded by 1 as $\mathcal{E}_{\mathcal{N}}^K(\sigma_k) \triangleq \left\{ \left( \mathcal{N}\left( \mu_i, \sigma_k^2 \right) \right)_{i=1}^K : \mu = (\mu_1, \ldots, \mu_K) \in \mathbb{R}^K \right\}$. Since $\Pi^\epsilon \subset \Pi$, we can have that

$$\mathcal{R}_T^{\text{minimax}}(\pi, \nu) \geq \inf_{\pi \in \Pi} \sup_{\nu \in \mathcal{E}_{\mathcal{N}}^K(\sigma_k)} \text{Reg}_T(\pi, \nu) \geq \Omega(\sqrt{KT})$$

where the last inequality is due to Theorem 15.2 in [55].

**Case 1: Uncontaminated case.** By the definition of minimax regret, we know that $\mathcal{R}_{\epsilon,\alpha}^{\text{minimax}} \geq \mathcal{R}_{\epsilon,0}^{\text{minimax}}$. Therefore, we first derive the lower bound of private bandits without contamination.

We consider two environments. In the first environment $\nu_1$, the optimal arm (denote by $a_1$) follows

$$r_{a_1} = \begin{cases} 1/\gamma & \text{with probability of } \frac{1}{2}\gamma^k \\ 0 & \text{with probability of } 1 - \frac{1}{2}\gamma^k \end{cases}$$

where $\gamma \in (0, 1]$. We can verify $\mathbb{E}[r_{a_1}] = \frac{1}{2}\gamma^{k-1}$ and $\mathbb{E}[r_{a_1}^k] = \frac{1}{2} \leq 1$. Any other sub-optimal arm $a \neq a_1$ in $\nu_1$ follows the same reward distribution

$$r_a = \begin{cases} 1/\gamma & \text{with probability of } \frac{3}{10}\gamma^k \\ 0 & \text{with probability of } 1 - \frac{3}{10}\gamma^k \end{cases}$$

We can verify $\mathbb{E}[r_a] = \frac{3}{10}\gamma^{k-1}$ and $\mathbb{E}[r_{a_1}^k] = \frac{3}{10} \leq 1$. Then the gap of means between the optimal arm and sub-optimal arm is $\Delta = \frac{1}{5}\gamma^{k-1}$.

For algorithm $\pi$ and instance $\nu_1$, we denote $i = \arg\min_{a \in \{2, \cdots, K\}} \mathbb{E}_{\pi, \nu_1}[N_a(T)]$. Thus, $\mathbb{E}_{\pi, \nu_1}[N_i(T)] \leq \frac{T}{K-1}$.

Now, consider another instance $\nu_2$ where $r_{a_1}, \cdots, r_{a_k}$ are the same as those in $\nu_1$ except the $i$-th arm such that

$$r_i' = \begin{cases} 1/\gamma & \text{with probability of } \frac{7}{10}\gamma^k \\ 0 & \text{with probability of } 1 - \frac{7}{10}\gamma^k \end{cases}$$

We can verify $\mathbb{E}[r_i'] = \frac{7}{10}\gamma^{k-1}$ and $\mathbb{E}[(r_i')^k] = \frac{7}{10} \leq 1$. Then in $\nu_2$, the arm $i$ is optimal.

Now by the classic regret decomposition, we obtain

$$\mathcal{R}_T(\pi, \nu_1) = (T - \mathbb{E}_{\pi, \nu_1}[N_1(T)])\Delta \geq \mathbb{P}_{\pi, \nu_1}^T\left[ N_1(T) \leq \frac{T}{2} \right]\frac{T\Delta}{2}.$$

$$\mathcal{R}_T(\pi, \nu_2) = \Delta\mathbb{E}_{\pi, \nu_2}[N_1(T)] + \sum_{a \notin \{1,i\}} 2\Delta\mathbb{E}_{\pi, \nu_2}[N_a(T)] \geq \mathbb{P}_{\pi, \nu_2}^T\left[ N_1(T) \geq \frac{T}{2} \right]\frac{T\Delta}{2}.$$

By applying the Bretagnolle–Huber inequality ([55], Theorem 14.2), we have

$$\mathcal{R}_T(\pi, \nu_1) + \mathcal{R}_T(\pi, \nu_2) \geq \frac{T\Delta}{2}\left( \mathbb{P}_{\pi, \nu_1}^T\left[ N_1(T) \leq \frac{T}{2} \right] + \mathbb{P}_{\pi, \nu_2}^T\left[ N_1(T) \geq \frac{T}{2} \right] \right).$$

$$\geq \frac{T\Delta}{4} \exp\left( -\text{KL}\left( \mathbb{P}_{\pi, \nu_1}^T \| \mathbb{P}_{\pi, \nu_2}^T \right) \right)$$

Based on Lemma C.1, we can get the upper bound of the KL-Divergence between the marginals.

$$\text{KL}\left( \mathbb{P}_{\pi, \nu_1}^T \| \mathbb{P}_{\pi, \nu_2}^T \right) \leq 6\epsilon\mathbb{E}_{\pi, \nu_1}\left[ \sum_{t=1}^T \text{TV}(r_{a_t} \| r_{a_t}') \right]$$

$$\leq 6\epsilon\mathbb{E}_{\pi, \nu_1}[N_i(T)]\text{TV}(r_i \| r_i')$$

since $\nu_1$ and $\nu_i$ only differ in the arm $i$.

Thus,

$$\mathcal{R}_T(\pi, \nu_1) + \mathcal{R}_T(\pi, \nu_2) \geq \frac{T\Delta}{4} \exp\left(-6\epsilon \mathbb{E}_{\pi, \nu_1}[N_i(T)] \cdot \frac{2}{5}\gamma^k\right)$$

$$\geq \frac{T\gamma^{k-1}}{20} \exp\left(-\frac{12 \cdot \epsilon T\gamma^k}{5(K-1)}\right).$$

Taking $\gamma = \left(\frac{K-1}{T\epsilon}\right)^{\frac{1}{k}}$, we get the result

$$\mathcal{R}_T(\pi, \nu_1) \geq \Omega\left(\left(\frac{K}{\epsilon}\right)^{\frac{k-1}{k}} T^{\frac{1}{k}}\right).$$

**Case 2: Contaminated case.** For $\alpha \neq 0$ and $\alpha \in (0, 1/2)$, we still consider the true distributions of arms are the same in above $\nu_1$ and $\nu_2$. In the first environment $\nu_1$, the optimal arm (denote by $a_1$) follows

$$r_{a_1} = \begin{cases} 1/\gamma & \text{with probability of } \frac{1}{2}\gamma^k \\ 0 & \text{with probability of } 1 - \frac{1}{2}\gamma^k \end{cases}$$

where $\gamma \in (0, 1]$. We can verify $\mathbb{E}[r_{a_1}] = \frac{1}{2}\gamma^{k-1}$ and $\mathbb{E}[r_{a_1}^k] = \frac{1}{2} \leq 1$. Any other sub-optimal arm $a \neq a_1$ in $\nu_1$ follows the same reward distribution

$$r_a = \begin{cases} 1/\gamma & \text{with probability of } \frac{3}{10}\gamma^k \\ 0 & \text{with probability of } 1 - \frac{3}{10}\gamma^k \end{cases}$$

We can verify $\mathbb{E}[r_a] = \frac{3}{10}\gamma^{k-1}$ and $\mathbb{E}[r_{a_1}^k] = \frac{3}{10} \leq 1$. Then the gap of means between the optimal arm and sub-optimal arm is $\Delta = \frac{1}{5}\gamma^{k-1}$.

And we denote the contaminated version of $\nu_1$ as $\tilde{\nu}_1$. For algorithm $\pi$ and instance $\tilde{\nu}_1$, we denote $i = \arg\min_{a \in \{2, \cdots, K\}} \mathbb{E}_{\pi, \tilde{\nu}_1}[N_a(T)]$. Thus, $\mathbb{E}_{\pi, \tilde{\nu}_1}[N_i(T)] \leq \frac{T}{K-1}$.

Now, consider another instance $\nu_2$ where $r_{a_1}, \cdots, r_{a_k}$ are the same as those in $\nu_1$ except the $i$-th arm such that

$$r_i' = \begin{cases} 1/\gamma & \text{with probability of } \frac{7}{10}\gamma^k \\ 0 & \text{with probability of } 1 - \frac{7}{10}\gamma^k \end{cases}$$

We can verify $\mathbb{E}[r_i'] = \frac{7}{10}\gamma^{k-1}$ and $\mathbb{E}[(r_i')^k] = \frac{7}{10} \leq 1$. Then in $\nu_2$, the arm $i$ is optimal.

Also, we denote the contaminated version of $\nu_2$ as $\tilde{\nu}_2$. Take $\gamma = \alpha^{\frac{1}{k}} \in (0, 1]$, since for any $a \in [K]$, $\text{TV}(r_a \| r_a') \leq \frac{2}{5}\gamma^k = \frac{2}{5}\alpha \leq \frac{\alpha}{1-\alpha}$, from Lemma C.2, we have for any arm $a \in [K]$, there exists distribution $G_a$ and $G_a'$ such that

$$(1-\alpha)r_a + \alpha G_a = (1-\alpha)r_a' + \alpha G_a'.$$

We consider $\tilde{\nu}_1 = \{x_a = (1-\alpha)r_a + \alpha G_a : a \in [K]\}$ and $\tilde{\nu}_2 = \{x_a' = (1-\alpha)r_a' + \alpha G_a' : a \in [K]\}$.

Now by the classic regret decomposition, we obtain

$$\mathcal{R}_T(\pi, \tilde{\nu}_1) = (T - \mathbb{E}_{\pi, \tilde{\nu}_1}[N_1(T)])\Delta \geq \mathbb{P}_{\pi, \tilde{\nu}_1}^T\left[N_1(T) \leq \frac{T}{2}\right]\frac{T\Delta}{2}.$$

$$\mathcal{R}_T(\pi, \tilde{\nu}_2) = \Delta\mathbb{E}_{\pi, \tilde{\nu}_2}[N_1(T)] + \sum_{a \notin \{1, i\}} 2\Delta\mathbb{E}_{\pi, \tilde{\nu}_2}[N_a(T)] \geq \mathbb{P}_{\pi, \tilde{\nu}_2}^T\left[N_1(T) \geq \frac{T}{2}\right]\frac{T\Delta}{2}.$$

By applying the Bretagnolle–Huber inequality ([55], Theorem 14.2), we have

$$\mathcal{R}_T(\pi, \tilde{\nu}_1) + \mathcal{R}_T(\pi, \tilde{\nu}_2) \geq \frac{T\Delta}{2}\left(\mathbb{P}_{\pi, \tilde{\nu}_1}^T\left[N_1(T) \leq \frac{T}{2}\right] + \mathbb{P}_{\pi, \tilde{\nu}_2}^T\left[N_1(T) \geq \frac{T}{2}\right]\right).$$

$$\geq \frac{T\Delta}{4}\exp\left(-\text{KL}\left(\mathbb{P}_{\pi, \tilde{\nu}_1}^T \| \mathbb{P}_{\pi, \tilde{\nu}_2}^T\right)\right)$$

Based on Lemma C.1, we can get the upper bound of the KL-Divergence between the marginals.

$$\mathrm{KL}\left(\mathbb{P}^T_{\pi,\tilde{\nu}_1}\|\mathbb{P}^T_{\pi,\tilde{\nu}_2}\right) \leq 6\epsilon\mathbb{E}_{\pi,\tilde{\nu}_1}\left[\sum_{t=1}^{T}\mathrm{TV}(x_{a_t}\|x'_{a_t})\right]$$

Since, $\mathrm{TV}(x_a\|x'_a) = 0$ for $\forall a \in [K]$, $\Delta = \frac{1}{5}\gamma^{k-1}$ and $\gamma = \alpha^{\frac{1}{k}}$. We obtain

$$\mathcal{R}_T(\pi,\tilde{\nu}_1) \geq \Omega(T\alpha^{1-\frac{1}{k}}).$$

Combine Gaussian case, case 1 and case 2, we have

$$\mathcal{R}_T = \Omega\left(\max\left\{\sqrt{KT}, \left(\frac{K}{\epsilon}\right)^{1-\frac{1}{k}}T^{\frac{1}{k}}, T\alpha^{1-\frac{1}{k}}\right\}\right).$$

$\square$

## D  Proofs of Section 6.1

**Proof of Theorem 6.1.** We denote the finite raw moments distribution for rewards by $P_k$, and denote $P_k$ under $\alpha$-Huber contamination by $P_{\alpha,k}$. Let $\hat{\mu} = \frac{1}{n}\sum_{\substack{i\in[n]\\X_i\sim P_{\alpha,k}}}X_i\mathbb{1}_{(|X_i|\leq M)}$ and $\mu = \mathbb{E}_{X_i\sim P_k}[X_i]$.

$$|\tilde{\mu}-\mu|$$

$$\leq\left|\mathrm{Lap}\left(\frac{2M}{n\epsilon}\right)\right| + |\hat{\mu}-\mu|$$

$$\leq\left|\mathrm{Lap}\left(\frac{2M}{n\epsilon}\right)\right| + \left|\frac{1}{n}\sum_{\substack{i\in[n]\\X_i\sim P_{\alpha,k}}}X_i\mathbb{1}_{(|X_i|\leq M)} - \mathbb{E}_{X_i\sim P_k}[X_i\mathbb{1}_{(|X_i|\leq M)}]\right| + |\mathbb{E}_{X_i\sim P_k}[X_i\mathbb{1}_{(|X_i|\leq M)}]-\mu|$$

$$=\left|\mathrm{Lap}\left(\frac{2M}{n\epsilon}\right)\right| + \left|\frac{1}{n}\sum_{\substack{i\in[n]\\X_i\sim P_{\alpha,k}}}X_i\mathbb{1}_{(|X_i|\leq M)} - \mathbb{E}_{X_i\sim P_k}[X_i\mathbb{1}_{(|X_i|\leq M)}]\right| + |\mathbb{E}_{X_i\sim P_k}[X_i\mathbb{1}_{(|X_i|>M)}]|$$

$$\overset{(a)}{\leq}\frac{2M\log(2/\delta)}{n\epsilon} + \left|\frac{1}{n}\sum_{\substack{i\in[n]\\X_i\sim P_{\alpha,k}}}X_i\mathbb{1}_{(|X_i|\leq M)} - \mathbb{E}_{X_i\sim P_k}[X_i\mathbb{1}_{(|X_i|\leq M)}]\right| + \mathbb{E}_{X_i\sim P_k}[|X_i|\mathbb{1}_{(|X_i|>M)}] \quad \text{w.p.} \quad 1-\frac{\delta}{2}$$

$$\overset{(b)}{\leq}\frac{2M\log(2/\delta)}{n\epsilon} + \left|\frac{1}{n}\sum_{\substack{i\in[n]\\X_i\sim P_{\alpha,k}}}X_i\mathbb{1}_{(|X_i|\leq M)} - \mathbb{E}_{X_i\sim P_k}[X_i\mathbb{1}_{(|X_i|\leq M)}]\right| + (\mathbb{E}_{X_i\sim P_k}[|X_i|^k])^{\frac{1}{k}}(\mathbb{P}_{X_i\sim P_k}(|X_i|>M))^{\frac{k-1}{k}}$$

$$\overset{(c)}{\leq}\frac{2M\log(2/\delta)}{n\epsilon} + \left|\frac{1}{n}\sum_{\substack{i\in[n]\\X_i\sim P_{\alpha,k}}}X_i\mathbb{1}_{(|X_i|\leq M)} - \mathbb{E}_{X_i\sim P_k}[X_i\mathbb{1}_{(|X_i|\leq M)}]\right| + \frac{1}{M^{k-1}}$$

where the inequality $(a)$ follows from Lemma B.5, $(b)$ is from Hölder's Inequality in Lemma B.7 and $(c)$ follows from Markov's inequality in Lemma B.3.

Now we focus on the upper bound of $\left|\frac{1}{n}\sum_{\substack{i\in[n]\\X_i\sim P_{\alpha,k}}}X_i\mathbb{1}_{(|X_i|\leq M)} - \mathbb{E}_{X_i\sim P_k}[X_i\mathbb{1}_{(|X_i|\leq M)}]\right|$. Let $N_G$ be the set of indices in $n$ samples distributed according to $G$, and $N_{P_k}$ be the set of indices in $n$ samples distributed according to $P_k$. Then

**Case 1: uncontaminated case** ($\alpha = 0$) Now, the only thing left is to upper bound

$$\left| \frac{1}{n} \sum_{\substack{i \in [n] \\ X_i \sim P_k}} X_i \mathbb{1}_{(|X_i| \leq M)} - \mathbb{E}_{X_i \sim P_k}[X_i \mathbb{1}_{(|X_i| \leq M)}] \right|.$$

For $X_i \sim P_k$, let $Y_i = X_i \mathbb{1}_{(|X_i| \leq M)}$, then $|Y_i| \leq M$ and $\mathrm{Var}(Y_i) = \mathbb{E}[Y_i^2] - (\mathbb{E}[Y_i])^2 \leq \mathbb{E}[Y_i^2] \leq \mathbb{E}_{X_i \sim P_k}[X_i^2] \leq 1$. Then, from Bernstein's inequality in Lemma B.8, we have with probability $1 - \delta/2$

$$\left| \frac{1}{n} \sum_{\substack{i \in [n] \\ X_i \sim P_k}} X_i \mathbb{1}_{(|X_i| \leq M)} - \mathbb{E}_{X_i \sim P_k}[X_i \mathbb{1}_{(|X_i| \leq M)}] \right| \leq \sqrt{\frac{2 \log(4/\delta)}{n}} + \frac{4M \log(4/\delta)}{3n}. \tag{3}$$

Then we get with probability at least $1 - \delta$,

$$|\tilde{\mu} - \mu| \leq \sqrt{\frac{2 \log(4/\delta)}{n}} + \frac{4M \log(4/\delta)}{3n} + \frac{2M \log(2/\delta)}{n\epsilon} + \frac{1}{M^{k-1}}.$$

For $\epsilon > 0$ and $\delta \in (0,1)$, we have with probability at least $1 - \delta$,

$$|\tilde{\mu} - \mu| \leq \sqrt{\frac{2 \log(4/\delta)}{n}} + \frac{4M \log(4/\delta)}{n\epsilon} + \frac{1}{M^{k-1}}.$$

Taking the truncation threshold $M = \left( \frac{n\epsilon}{4 \log(4/\delta)} \right)^{\frac{1}{k}}$, we have

$$|\tilde{\mu} - \mu| \leq \sqrt{\frac{2 \log(4/\delta)}{n}} + 2 \left( \frac{4 \log(4/\delta)}{n\epsilon} \right)^{1 - \frac{1}{k}}.$$

**Case 2: contaminated case** ($\alpha \in (0, \frac{1}{2})$)

$$\left| \frac{1}{n} \sum_{\substack{i \in [n] \\ X_i \sim P_{\alpha,k}}} X_i \mathbb{1}_{(|X_i| \leq M)} - \mathbb{E}_{X_i \sim P_k}[X_i \mathbb{1}_{(|X_i| \leq M)}] \right|$$

$$= \left| \frac{1}{n} \sum_{i \in N_G} X_i \mathbb{1}_{(|X_i| \leq M)} + \frac{1}{n} \sum_{i \in N_{P_k}} X_i \mathbb{1}_{(|X_i| \leq M)} - \mathbb{E}_{X_i \sim P_k}[X_i \mathbb{1}_{(|X_i| \leq M)}] \right|$$

$$\leq \underbrace{\left| \frac{1}{n} \sum_{i \in N_G} X_i \mathbb{1}_{(|X_i| \leq M)} \right|}_{T_1} + \underbrace{\left| \frac{1}{n} \sum_{i \in N_{P_k}} X_i \mathbb{1}_{(|X_i| \leq M)} - \mathbb{E}_{X_i \sim P_k}[X_i \mathbb{1}_{(|X_i| \leq M)}] \right|}_{T_2}.$$

To control $T_1$, we can write it as

$$T_1 = \left| \frac{1}{n} \sum_{i \in N_G} X_i \mathbb{1}_{(|X_i| \leq M)} \right|$$

$$\leq \frac{1}{n} \sum_{i \in N_G} |X_i| \, \mathbb{1}_{(|X_i| \leq M)}$$

$$\leq \frac{|N_G|}{n} M.$$

Then $\frac{|N_G|}{n}$ can be treat as a mean estimation of Bernoulli distribution $Ber(\alpha)$. Then based on Bernstein's inequality in Lemma B.8, we get with probability $1 - \delta/4$,

$$\left| \frac{|N_G|}{n} - \alpha \right| \leq \sqrt{\frac{2\alpha(1-\alpha) \log(8/\delta)}{n}} + \frac{2 \log(8/\delta)}{3n} \leq \sqrt{\frac{2\alpha_1(1-\alpha_1) \log(8/\delta)}{n}} + \frac{2 \log(8/\delta)}{3n}$$

for $\alpha \leq \alpha_1 \in (0, 1/2)$.

Thus,
$$T_1 \leq \left( \alpha_1 + \sqrt{\frac{2\alpha_1 \log(8/\delta)}{n}} + \frac{2\log(8/\delta)}{3n} \right) M. \quad \text{with probability} 1 - \delta/4$$

When $n \geq \frac{\log(8/\delta)}{\alpha_1}$, we have
$$T_1 \leq 4\alpha_1 M.$$

To bound $T_2$, we have
$$\left| \frac{1}{n} \sum_{i \in N_{P_k}} X_i \mathbb{1}_{(|X_i| \leq M)} - \mathbb{E}_{X_i \sim P_k}[X_i \mathbb{1}_{(|X_i| \leq M)}] \right|$$

$$= \left| \frac{1}{n} \sum_{\substack{i \in N_G \cup N_{P_k} \\ X_i \sim P_k}} X_i \mathbb{1}_{(|X_i| \leq M)} - \frac{1}{n} \sum_{\substack{i \in N_G \\ X_i \sim P_k}} X_i \mathbb{1}_{(|X_i| \leq M)} - \mathbb{E}_{X_i \sim P_k}[X_i \mathbb{1}_{(|X_i| \leq M)}] \right|$$

$$\leq \left| \frac{1}{n} \sum_{\substack{i \in N_G \\ X_i \sim P_k}} X_i \mathbb{1}_{(|X_i| \leq M)} \right| + \left| \frac{1}{n} \sum_{\substack{i \in [n] \\ X_i \sim P_k}} X_i \mathbb{1}_{(|X_i| \leq M)} - \mathbb{E}_{X_i \sim P_k}[X_i \mathbb{1}_{(|X_i| \leq M)}] \right|$$

$$\leq 4\alpha_1 M + \sqrt{\frac{2\log(16/\delta)}{n}} + \frac{4M\log(16/\delta)}{3n} \quad \text{w.p.} \quad 1 - \delta/4$$

where the last inequality is based on the similar analysis of $T_1$ and the inequality of (3).

Put everything together, we have with probability at least $1 - \delta$,
$$|\tilde{\mu} - \mu| \leq \sqrt{\frac{2\log(16/\delta)}{n}} + \frac{4M\log(16/\delta)}{3n} + 8\alpha_1 M + \frac{2M\log(2/\delta)}{n\epsilon} + \frac{1}{M^{k-1}}.$$

Thus, for $\epsilon > 0$, we have
$$|\tilde{\mu} - \mu| \leq \sqrt{\frac{2\log(16/\delta)}{n}} + \frac{4M\log(16/\delta)}{n\epsilon} + \frac{1}{M^{k-1}} + 8\alpha_1 M.$$

Taking $M = \min \left\{ \left( \frac{n\epsilon}{4\log(16/\delta)} \right)^{\frac{1}{k}}, (8\alpha_1)^{-\frac{1}{k}} \right\}$, we have
$$|\tilde{\mu} - \mu| \leq \sqrt{\frac{2\log(16/\delta)}{n}} + 2\left( \frac{4\log(16/\delta)}{n\epsilon} \right)^{1-\frac{1}{k}} + 2(8\alpha_1)^{1-\frac{1}{k}}.$$

$\square$

**Proof of Theorem 6.2.** Let $\tau_0$ be the maximal epoch such that $B_\tau < \frac{\log(16|\mathcal{S}|\tau^2/\delta)}{\alpha_1}$.

For all epoch $\tau \leq \tau_0$, the batch size is less than $2^{\tau_0}$. Since batch size doubles, until epoch $\tau_0$, we have the number of pulls for each arm $a \in [K]$ is less than $2 \cdot 2^{\tau_0} \leq 2\frac{\log(16|\mathcal{S}|\tau_0^2/\delta)}{\alpha_1}$. Then the regret has to suffer $\frac{2\log(16|\mathcal{S}|\tau_0^2/\delta)}{\alpha_1} \Delta_a$ for each $a \in [K]$.

For $\tau > \tau_0$, $B_\tau \geq \frac{\log(16|\mathcal{S}|\tau^2/\delta)}{\alpha_1}$. For each $a \in \mathcal{S}$, from Theorem 6.1, we have with probability at least $1 - \frac{\delta}{2|\mathcal{S}|\tau^2}$,
$$|\tilde{\mu}_a - \mu_a| \leq \beta_\tau.$$

Given an epoch $\tau > \tau_0$, we denote by $\mathcal{E}_\tau$ the event where for all $a \in \mathcal{S}$ it holds that $|\tilde{\mu}_a - \mu_a| \leq \beta_\tau$. and denote $\mathcal{E} = \cup_{\tau > \tau_0} \mathcal{E}_\tau$. By taking union bound, we have
$$\mathbb{P}(\mathcal{E}_\tau) \geq 1 - \frac{\delta}{2\tau^2},$$

and

$$\mathbb{P}(\mathcal{E}) \geq 1 - \frac{\delta}{2}\left(\sum_{\tau > \tau_0} \tau^{-2}\right) \geq 1 - \delta.$$

In the following, we condition on the good event $\mathcal{E}$. We first show that the optimal arm $a^*$ is never eliminated. For any epoch $\tau > \tau_0$, let $a_\tau = \arg\max_{a \in \mathcal{S}} \tilde{\mu}_a$. Since

$$(\tilde{\mu}_{a_\tau} - \tilde{\mu}_{a^*}) + \Delta_{a_\tau} = |(\tilde{\mu}_{a_\tau} - \tilde{\mu}_{a^*}) + \Delta_{a_\tau}| \leq |\tilde{\mu}_{a_\tau} - \mu_{a_\tau}| + |\tilde{\mu}_{a^*} - \mu_{a^*}| \leq 2\beta_\tau,$$

it is easy to see that the algorithm doesn't eliminate $a^*$.

Then, we show that at the end of epoch $\tau > \tau_0$, all arms such that $\Delta_a \geq 4\beta_\tau$ will be eliminated. To show this, we have that under good event $\mathcal{E}$,

$$\tilde{\mu}_a + \beta_\tau \leq \mu_a + 2\beta_\tau < \mu_{a^*} - 4\beta_\tau + 2\beta_\tau \leq \tilde{\mu}_{a^*} - \beta_\tau \leq \tilde{\mu}_{a_\tau} - \beta_\tau$$

which implies that arm $a$ will be eliminated by the rule. Thus, for each sub-optimal arm $a$, let $\tau(a)$ be the last epoch that arm $a$ is not eliminated. By the above result, we have

$$\Delta_a \leq 4\beta_{\tau(a)} = 4\sqrt{\frac{2\log(16|\mathcal{S}|\tau(a)^2/\delta)}{B_{\tau(a)}}} + 8\left(\frac{4\log(16|\mathcal{S}|\tau(a)^2/\delta)}{B_{\tau(a)}\epsilon}\right)^{1-\frac{1}{k}} + 8(8\alpha_1)^{1-\frac{1}{k}}.$$

We divide the arms $a \in [K]$ into two groups: $\mathcal{G}_1 = \{a \in [K] : 16(8\alpha_1)^{1-\frac{1}{k}} \leq \Delta_a\}$ and $\mathcal{G}_2 = \{a \in [K] : 16(8\alpha_1)^{1-\frac{1}{k}} \geq \Delta_a\}$.

**Group 1:** Now, for all arm $a \in \mathcal{G}_1$, we have

$$\Delta_a \leq 8\sqrt{\frac{2\log(16|\mathcal{S}|\tau(a)^2/\delta)}{B_{\tau(a)}}} + 16\left(\frac{4\log(16|\mathcal{S}|\tau(a)^2/\delta)}{B_{\tau(a)}\epsilon}\right)^{1-\frac{1}{k}}.$$

Hence, we have

$$B_{\tau(a)} \leq \max\left\{\frac{128\log(16|\mathcal{S}|\tau(a)^2/\delta)}{\Delta_a^2}, \frac{4\log(16|\mathcal{S}|\tau(a)^2/\delta)}{\epsilon}\left(\frac{16}{\Delta_a}\right)^{\frac{k}{k-1}}, \frac{\log(16|\mathcal{S}|\tau_0^2/\delta)}{\alpha_1}\right\}.$$

Since $|\mathcal{S}| \leq K$ and $2^\tau \leq T$ for any $\tau$. Thus,

$$B_{\tau(a)} \leq \max\left\{\frac{128\log(16K\log^2 T/\delta)}{\Delta_a^2}, \frac{4\log(16K\log^2 T/\delta)}{\epsilon}\left(\frac{16}{\Delta_a}\right)^{\frac{k}{k-1}}, \frac{\log(16K\log^2 T/\delta)}{\alpha_1}\right\},$$

Since the batch size doubles, we have $N_a(T) \leq 2B_{\tau(a)}$ for each sub-optimal arm $a$. Therefore, for all arm $a \in \mathcal{G}_1$,

$$\mathcal{R}_T = \sum_{a \in \mathcal{G}_1} N_a(T)\Delta_a \leq 2B_{\tau(a)}\Delta_a.$$

Let $\eta$ be a number in $(0, 1)$. For all arms $a \in \mathcal{G}_1$ with $\Delta_a \leq \eta$, the regret incurred by pulling these arms is upper bounded by $T\eta$. For any arm $a \in \mathcal{G}_1$ with $\Delta_a > \eta$, choose $\delta = \frac{1}{T}$ and assume $T \geq K$, then the expected regret incurred by pulling arm $a$ is upper bounded by

$$\mathbb{E}\left[\sum_{a \in \mathcal{G}_1, \Delta_a > \eta} \Delta_a N_a(T)\right] \leq \mathbb{P}(\bar{\mathcal{E}}) \cdot T + O\left(\sum_{a \in \mathcal{G}_1, \Delta_a > \eta}\left\{\frac{\log T}{\Delta_a} + \frac{\log T}{\epsilon}\left(\frac{1}{\Delta_a}\right)^{\frac{1}{k-1}} + \frac{\log T}{\alpha_1}\Delta_a\right\}\right)$$

$$\leq O\left(\frac{K\log T}{\eta} + \frac{K\log T}{\epsilon\eta^{\frac{1}{k-1}}} + \frac{K\log T}{\alpha_1}\right)$$

where the last term in the last inequality is based on following result: from the heavy-tailed assumption for rewards distributions in (1), we have for any $a \in [K]$, $|\mu_a| \leq \mathbb{E}_{r_a \sim P_k}|r_a| \leq \mathbb{E}_{r_a \sim P_k}|r_a|^k \leq 1$, so $\Delta_a = \mu^* - \mu_a \leq 2$.

Thus the regret from group 1 is at most

$$T\eta + O\left(\frac{K\log T}{\eta} + \frac{K\log T}{\epsilon\eta^{\frac{1}{k-1}}} + \frac{K\log T}{\alpha_1}\right).$$

Taking $\eta = \max\left\{\sqrt{\frac{K\log T}{T}}, \left(\frac{K\log T}{T\epsilon}\right)^{\frac{k-1}{k}}\right\}$, the regret from group 1 is at most

$$O\left(\sqrt{KT\log T} + \left(\frac{K\log T}{\epsilon}\right)^{\frac{k-1}{k}} T^{\frac{1}{k}} + \frac{K\log T}{\alpha_1}\right)$$

**Group 2:** For all other arms $a \in \mathcal{G}_2$, we have the total regret is at most $O(T\Delta_a) = O(T\alpha_1^{1-\frac{1}{k}})$.

Combine the two groups, choose $\delta = \frac{1}{T}$ and assume $T \geq K$, we have the that the expected regret satisfies,

$$\mathcal{R}_T \leq O\left(\sqrt{KT\log T} + \left(\frac{K\log T}{\epsilon}\right)^{\frac{k-1}{k}} T^{\frac{1}{k}} + \frac{K\log T}{\alpha_1} + T\alpha_1^{1-\frac{1}{k}}\right)$$

We also give privacy guarantee for the algorithm. Based on Laplacian mechanism in Definition 3.5 and Post-processing in Lemma B.1, we can get that Algorithm 1 is $\epsilon$-DP. $\qquad\square$

# E   Proofs of Section 6.2

**Proof of Theorem 6.5.** **Step 1:** we will show that with high probability $1 - \delta/2$, $|J - \mu| \leq 2r$.

To this end, we first study the private histogram. Note $\mathbb{E}[\mathbb{1}(X_i \in B_j)] = P_{\alpha,k}(B_j)$, then

$$\mathbb{P}\left(|\tilde{p}_j - P_{\alpha,k}(B_j)| > t\right) = \mathbb{P}\left(\left|\frac{\sum_{i=1}^n \mathbb{1}(X_i \in B_j)}{n} + \mathrm{Lap}\left(\frac{2}{n\epsilon}\right) - P_{\alpha,k}(B_j)\right| > t\right)$$

$$\leq \mathbb{P}\left(\left|\frac{\sum_{i=1}^n \mathbb{1}(X_i \in B_j)}{n} - P_{\alpha,k}(B_j)\right| > t/2\right) + \mathbb{P}\left(\left|\mathrm{Lap}\left(\frac{2}{n\epsilon}\right)\right| > t/2\right)$$

$$\leq 2\exp\left(-\frac{nt^2}{2}\right) + \exp\left(-\frac{n\epsilon t}{4}\right),$$

where the last inequality is from Lemma B.6 and Lemma B.5. By a union bound over $j$, we further have

$$\mathbb{P}\left(\max_{j\in\mathcal{J}} |\tilde{p}_j - P_{\alpha,k}(B_j)| > t\right) \leq \frac{2D}{r}\left(2\exp\left(-\frac{nt^2}{2}\right) + \exp\left(-\frac{n\epsilon t}{4}\right)\right)$$

$$\leq 2D\left(2\exp\left(-\frac{nt^2}{2}\right) + \exp\left(-\frac{n\epsilon t}{4}\right)\right)$$

Thus, we have with probability $1 - \delta/2$,

$$\max_{j\in\mathcal{J}} |\tilde{p}_j - P_{\alpha,k}(B_j)| \leq \max\left\{\sqrt{\frac{2\ln\frac{16D}{\delta}}{n}}, \frac{4\ln\frac{16D}{\delta}}{n\epsilon}\right\} := C_1.$$

In the following, we condition on the above event. Next, by Chebyshev's inequality in Lemma B.4 and the assumption of $\mathcal{P}_k$ that $k$-th central moment is less than 1, we have

$$\mathbb{P}_{X\sim\mathcal{P}_{\alpha,k}}\left(|X-\mu| \geq r\right) \leq \alpha\mathbb{P}_{X\sim\mathcal{G}}(|X-\mu| \geq r) + (1-\alpha)\mathbb{P}_{X\sim\mathcal{P}_k}(|X-\mu| \geq r)$$

$$\leq \alpha + (1-\alpha)(1/r)^k := C_2 \qquad\qquad (4)$$

Let $j^*$ is the index of the bin containing the true mean $\mu$ and we consider three consecutive intervals $A_{j^*} = B_{j^*-1} \cup B_{j^*} \cup B_{j^*+1}$

$$
\begin{aligned}
P_{\alpha,k}(A_{j^*}) &= P_{\alpha,k}(B_{j^*-1}) + P_{\alpha,k}(B_{j^*}) + P_{\alpha,k}(B_{j^*+1}) \\
&\geq P_{\alpha,k}\left((\mu - r, \mu + r)\right) \\
&\geq 1 - C_2.
\end{aligned}
$$

where the first inequality is from inequality (4). Now, for any $j \notin \{j^* - 1, j^*, j^* + 1\}$, we have when $D \geq 2r$

$$
\tilde{p}_j \leq P_{\alpha,k}(B_j) + C_1 \leq 1 - P_{\alpha,k}(A_{j^*}) + C_1 \leq C_2 + C_1.
$$

On the other hand, since $P_{\alpha,k}(A_{j^*}) \geq 1 - C_2$, there must exist some $j \in \{j^* - 1, j^*, j^* + 1\}$ such that $P_{\alpha,k}(B_j) \geq \frac{1-C_2}{3}$. Therefore, for this $j$, we have

$$
\tilde{p}_j \geq P_{\alpha,k}(B_j) - C_1 \geq \frac{1 - C_2}{3} - C_1.
$$

Therefore, if $n$ (depending on $\alpha, \epsilon, r$) such that $\frac{1-C_2}{3} - C_1 > C_2 + C_1$, the true mean $\mu$ is in the bin chosen by line 3 in Algorithm 3 or it's neighboring bin, which implies that with probability at least $1 - \delta/2$, $|J - \mu| \leq 2r$.

**Step 2:** Utilizing the above result, we aim to show that truncation can handle heavy-tail, privacy and robustness in the concentration.

$$
\begin{aligned}
|\tilde{\mu} - \mu| &= |J + \frac{1}{n} \sum_{i=n+1}^{2n} (X_i - J)\mathbb{1}(|X_i - J| \leq M) + \mathrm{Lap}\left(\frac{2M}{n\epsilon}\right) - \mu| \\
&= \left| \frac{1}{n} \sum_{i=n+1}^{2n} (X_i - J)\mathbb{1}(|X_i - J| \leq M) + \mathrm{Lap}\left(\frac{2M}{n\epsilon}\right) \right. \\
&\quad \left. + \frac{1}{n} \sum_{i=n+1}^{2n} (J - \mu)\left\{\mathbb{1}(|X_i - J| \leq M) + \mathbb{1}(|X_i - J| > M)\right\} \right| \\
&= \left| \frac{1}{n} \sum_{i=n+1}^{2n} (X_i - J + J - \mu)\mathbb{1}(|X_i - J| \leq M) + \mathrm{Lap}\left(\frac{2M}{n\epsilon}\right) + \frac{1}{n} \sum_{i=n+1}^{2n} (J - \mu)\mathbb{1}(|X_i - J| > M) \right| \\
&\leq \left| \frac{1}{n} \sum_{i=n+1}^{2n} (X_i - \mu)\mathbb{1}(|X_i - J| \leq M) \right| + \left| \mathrm{Lap}\left(\frac{2M}{n\epsilon}\right) \right| + \left| \frac{1}{n} \sum_{i=n+1}^{2n} (J - \mu)\mathbb{1}(|X_i - J| > M) \right|
\end{aligned}
$$

We first focus on the first term in the right hand of the last inequality. Let $N_G$ be the set of indices in $n$ samples distributed according to $G \in \mathcal{G}$, and $N_{P_k}$ be the set of indices in $n$ samples distributed according to $P_k \in \mathcal{P}_k^c$. Then, we have

$$
\begin{aligned}
&\left| \frac{1}{n} \sum_{i=n+1}^{2n} (X_i - \mu)\mathbb{1}(|X_i - J| \leq M) \right| \\
&\leq \underbrace{\left| \frac{1}{n} \sum_{i \in N_G} (X_i - \mu)\mathbb{1}(|X_i - J| \leq M) \right|}_{T_1} + \underbrace{\left| \frac{1}{n} \sum_{i \in N_{P_k}} (X_i - \mu)\mathbb{1}(|X_i - J| \leq M) \right|}_{T_2}.
\end{aligned}
$$

To control $T_1$, we can write it as

$$T_1 = \left| \frac{1}{n} \sum_{i \in N_G} (X_i - \mu) \mathbb{1}(|X_i - J| \leq M) \right|$$

$$\leq \frac{1}{n} \sum_{i \in N_G} |(X_i - \mu)| \mathbb{1}(|X_i - J| \leq M)$$

$$\leq \frac{1}{n} \sum_{i \in N_G} |(X_i - \mu)| \mathbb{1}(|X_i - \mu| \leq M + 2r)$$

$$\leq \frac{|N_G|}{n} (M + 2r).$$

Then $\frac{|N_G|}{n}$ can be treated as a mean estimation of Bernoulli distribution $Ber(\alpha)$. Then based on Bernstein's inequality in Lemma B.8, we get with probability $1 - \delta/8$,

$$\left| \frac{|N_G|}{n} - \alpha \right| \leq \sqrt{\frac{2\alpha(1 - \alpha) \log(16/\delta)}{n}} + \frac{2 \log(16/\delta)}{3n} \leq \sqrt{\frac{2\alpha_1(1 - \alpha_1) \log(16/\delta)}{n}} + \frac{2 \log(16/\delta)}{3n}$$

for $\alpha \leq \alpha_1 \in (0, 1/2)$.

Thus,

$$T_1 \leq \left( \alpha_1 + \sqrt{\frac{2\alpha_1 \log(16/\delta)}{n}} + \frac{2 \log(16/\delta)}{3n} \right) (M + 2r), \quad \text{with probability} \quad 1 - \delta/8.$$

Thus, if $n$ satisfies $\sqrt{\frac{2\alpha_1 \log(16/\delta)}{n}} + \frac{2 \log(16/\delta)}{3n} = O(\alpha_1)$, then we have $T_1 = O(\alpha_1(M + 2r))$ Now, we bound $T_2$,

$$T_2 = \left| \frac{1}{n} \sum_{\substack{i \in N_G \cup N_{P_k} \\ X_i \sim P_k}} (X_i - \mu) \mathbb{1}(|X_i - J| \leq M) - \frac{1}{n} \sum_{\substack{i \in N_G \\ X_i \sim P_k}} (X_i - \mu) \mathbb{1}(|X_i - J| \leq M) \right|$$

$$\leq \left| \frac{1}{n} \sum_{\substack{i \in N_G \cup N_{P_k} \\ X_i \sim P_k}} (X_i - \mu) \mathbb{1}(|X_i - J| \leq M) \right| + \left| \frac{1}{n} \sum_{\substack{i \in N_G \\ X_i \sim P_k}} (X_i - \mu) \mathbb{1}(|X_i - J| \leq M) \right|$$

$$\leq \left| \frac{1}{n} \sum_{\substack{i \in [n] \\ X_i \sim P_k}} (X_i - \mu) \mathbb{1}(|X_i - J| \leq M) \right| + T_1.$$

Now we focus on the upper bound of $\left| \frac{1}{n} \sum_{\substack{i \in [n] \\ X_i \sim P_k}} (X_i - \mu) \mathbb{1}(|X_i - J| \leq M) \right|$. With probability $1 - \delta/8$,

$$
\left| \frac{1}{n} \sum_{\substack{i \in [n] \\ X_i \sim P_k}} (X_i - \mu) \mathbb{1}(|X_i - J| \leq M) \right|
$$

$$
\leq \left| \frac{1}{n} \sum_{\substack{i \in [n] \\ X_i \sim P_k}} (X_i - \mu) \mathbb{1}(|X_i - J| \leq M) - \mathbb{E}[(X_1 - \mu) \mathbb{1}(|X_1 - J| \leq M)] \right|
$$

$$
+ |\mathbb{E}[(X_1 - \mu) \mathbb{1}(|X_1 - J| \leq M)] - \mathbb{E}[(X_1 - \mu)]|
$$

$$
\leq \sqrt{\frac{2 \log(16/\delta)}{n}} + \frac{4(M + 2r) \log(16/\delta)}{3n} + |\mathbb{E}[(X_i - \mu) \mathbb{1}(|X_i - J| \geq M)]|
$$

$$
\leq \sqrt{\frac{2 \log(16/\delta)}{n}} + \frac{4(M + 2r) \log(16/\delta)}{3n} + \left( \mathbb{E}[|X_i - \mu|^k] \right)^{\frac{1}{k}} \left( \mathbb{P}(|X_i - \mu| \geq M - 2r) \right)^{\frac{k-1}{k}} ]
$$

$$
\leq \sqrt{\frac{2 \log(16/\delta)}{n}} + \frac{4(M + 2r) \log(16/\delta)}{3n} + \frac{1}{(M - 2r)^{k-1}}
$$

$$
\leq \sqrt{\frac{2 \log(16/\delta)}{n}} + \frac{4(M + 2r) \log(16/\delta)}{3n} + \left( \frac{2}{M} \right)^{k-1}
$$

where the last inequality follows from $M \geq 4r$, the third inequality follows from Hölder's Inequality in Lemma B.7 and the second inequality follows from Bernstein inequality in Lemma B.8. That is, let

$$
Y_i = (X_i - \mu) \mathbb{1}(|X_i - J| \leq M),
$$

then

$$
\begin{aligned}
|Y_i - \mathbb{E}[Y_i]| &\leq |Y_i| + |\mathbb{E}[Y_i]| \\
&\leq |X_i - \mu| \mathbb{1}(|X_i - \mu| \leq M + 2r) + \mathbb{E}[|X_i - \mu| \mathbb{1}(|X_i - \mu| \leq M + 2r)] \\
&\leq 2(M + 2r)
\end{aligned}
$$

and

$$
\begin{aligned}
\mathrm{Var}(Y_i - \mathbb{E}[Y_i]) = \mathbb{E}(Y_i - \mathbb{E}[Y_i])^2 &\leq \mathbb{E}[Y_i^2] \\
&\leq \mathbb{E}_{X_i \sim P_k}[(X_i - \mu)^2 \mathbb{1}(|X_i - \mu| \leq M + 2r)] \\
&\leq \mathbb{E}_{X_i \sim P_k}[(X_i - \mu)^2] \leq 1.
\end{aligned}
$$

Therefore, with probability $1 - 3\delta/8$,

$$
T_2 \leq \sqrt{\frac{2 \log(16/\delta)}{n}} + \frac{4(M + 2r) \log(16/\delta)}{3n} + \left( \frac{2}{M} \right)^{k-1} + T_1.
$$

Now, we focus on the upper bound of $T_3 := \left| \frac{1}{n} \sum_{i=n+1}^{2n} (J - \mu) \mathbb{1}(|X_i - J| > M) \right|$.

$$
\left| \frac{1}{n} \sum_{i=n+1}^{2n} (J - \mu) \mathbb{1}(|X_i - J| > M) \right|
$$

$$
\leq \frac{1}{n} \sum_{i=n+1}^{2n} |J - \mu| \, \mathbb{1}(|X_i - J| > M)
$$

$$
\leq 2r \frac{\sum_{i=n+1}^{2n} \mathbb{1}(|X_i - J| > M)}{n}
$$

$$
\leq 2r \frac{\sum_{i=n+1}^{2n} \mathbb{1}(|X_i - \mu| > M - 2r)}{n}
$$

where

$$\mathbb{E}_{X_i \sim P_{k,\alpha}}[\mathbb{1}(|X_i - \mu| > M - 2r)] = \mathbb{P}_{X_i \sim P_{k,\alpha}}(|X_i - \mu| > M - 2r)$$

$$\leq \alpha + (1 - \alpha)\mathbb{P}_{X_i \sim P_k}(|X_i - \mu| > M - 2r)$$

$$\leq \alpha + \frac{1}{(M - 2r)^k} \leq \alpha_1 + \left(\frac{2}{M}\right)^k$$

By Hoeffding's inequality, we have with probability $1 - \delta/8$,

$$\frac{\sum_{i=n+1}^{2n} \mathbb{1}(|X_i - \mu| > M - 2r)}{n} \leq \mathbb{P}_{X_i \sim P_{k,\alpha}}(|X_i - \mu| > M - 2r) + \sqrt{\frac{\log(16/\delta)}{2n}}.$$

Thus, we have

$$T_3 = \left| \frac{1}{n} \sum_{i=n+1}^{2n} (J - \mu)\mathbb{1}(|X_i - J| > M) \right| \leq 2r \left( \alpha_1 + \left(\frac{2}{M}\right)^k + \sqrt{\frac{\log(16/\delta)}{2n}} \right).$$

Putting everything together, we have

$$|\tilde{\mu} - \mu| = O\left( \left( \alpha_1 + \sqrt{\frac{2\alpha_1 \log(16/\delta)}{n}} + \frac{2\log(16/\delta)}{3n} \right)(M + 2r) \right)$$

$$+ O\left( \sqrt{\frac{2\log(16/\delta)}{n}} + \frac{4(M + 2r)\log(16/\delta)}{3n} + \left(\frac{2}{M}\right)^{k-1} \right)$$

$$+ O\left( 2r \left( \alpha_1 + \left(\frac{2}{M}\right)^k + \sqrt{\frac{\log(16/\delta)}{2n}} \right) \right)$$

$$+ O\left( \frac{M\log(1/\delta)}{n\epsilon} \right)$$

**Case I: $\alpha = 0$, Uncontaminated concentration.** We want to show that our concentration is better than medians-of-mean in [38] (Theorem 3.5). That is, we are additive for their third term therein (i.e., $\log(D) + \log(1/\delta)$), while they are multiplicative.

In this case, our $C_2 = (1/r)^k$, and by our first condition on $n$, it need to satisfy $6C_1 + 4C_2 < 1$. This implies that $C_2 < 1/4$. Thus, setting $r = 10^{1/k}$ is sufficient. Hence, we have $C_1 < 0.1$, which requires $n$ to satisfy $n \geq 200 \log(16D/\delta)$ and $n \geq 20 \log(16D/\delta)/\epsilon$. We can safely set $n \geq 200 \log(16D/\delta)/\epsilon$.

In the case of $\alpha = 0$, $T_1$ is not a problem, which only introduces another $O(M\log(1/\delta)/n)$. $T_3$ is also not a problem which is dominated by $O((2/M)^{k-1} + \sqrt{\log(1/\delta)}/\sqrt{n})$

Let's summarize all the values: when $\alpha = 0$, $r = 10^{1/k}$ and $n \geq 200 \log(16D/\delta)/\epsilon$, we have

$$|\tilde{\mu} - \mu| = O\left( \sqrt{\frac{2\log(16/\delta)}{n}} + \frac{M\log(16/\delta)}{3n} + \left(\frac{2}{M}\right)^{k-1} \right)$$

$$+ O\left( \frac{M\log(1/\delta)}{n\epsilon} \right)$$

$$= O\left( \sqrt{\frac{2\log(16/\delta)}{n}} + \frac{M\log(16/\delta)}{\epsilon n} + \left(\frac{2}{M}\right)^{k-1} \right)$$

Now, we need to choose $M$ to minimize the above while satisfying $M \geq 4r$. By standard choice, we set $M = 4\left(\frac{n\epsilon}{\log(1/\delta)}\right)^{1/k}$, which satisfies $M \geq 4r$ when $n \geq \frac{10\log(1/\delta)}{\epsilon}$.

**Case II: $\alpha > 0$ and $\alpha \in (0, 1/2)$. Contaminated concentration.** We want to minimize the term $\mathcal{T}(\alpha, \epsilon)$ while maximizing the possible range of $\alpha$.

In this case, $C_2 = \alpha + (1-\alpha)(1/r)^k$ and again we need to satisfy that $6C_1 + 4C_2 < 1$, which first implies that $\alpha$ needs to be $\alpha < 1/4$. Setting $r = \iota^{1/k}$, we have $C_2 = \alpha + \frac{1}{\iota}(1-\alpha)$, which needs to be less than $1/4$. Let's set $\iota = \frac{1-\alpha}{0.249-\alpha}$ (hence $\alpha < 0.249$), we have there exists an absolute constant $c_1$ such that when $n \geq c_1 \log(16D/\delta)/\epsilon$, we guarantee $6C_1 + 4C_2 < 1$.

Now, we turn to $T_1$. If $n \geq \frac{\log(16/\delta)}{\alpha_1}$ and $M \geq 4r$, we have $T_1 = O(\alpha_1 M)$.

For $T_3$, we have

$$T_3 = 2r\left(\alpha_1 + \left(\frac{2}{M}\right)^k + \sqrt{\frac{\log(16/\delta)}{2n}}\right)$$

One simple way is to set $n \geq \log(16/\delta)/\alpha_1^2$. Then, we have $T_3 = O(\alpha_1 M + (1/M)^{k-1})$.

Let's summarize it. For any $\alpha \in (0, 0.249)$, setting $r = \left(\frac{1-\alpha}{0.249-\alpha}\right)^{1/k}$. Then, for all $n \geq \max\{c_1 \log(16D/\delta)/\epsilon, \log(16/\delta)/\alpha_1^2\}$, we have

$$|\tilde{\mu} - \mu| = O\left(\sqrt{\frac{2\log(16/\delta)}{n}} + \frac{M\log(16/\delta)}{\epsilon n} + \left(\frac{2}{M}\right)^{k-1} + \alpha_1 M\right)$$

Now, we need to choose $M$ to minimize the above while satisfying $M \geq 4r$. By standard choice, we set $M = \min\{4\left(\frac{n\epsilon}{\log(1/\delta)}\right)^{1/k}, 4\alpha_1^{-1/k}\}$, which satisfies $M \geq 4r$ when $n$ and $\alpha$ satisfy

$$n \geq \frac{\iota\log(1/\delta)}{\epsilon} \quad \text{and} \quad \frac{1}{\alpha} \geq \iota,$$

where recall that $\iota = \frac{1-\alpha}{0.249-\alpha}$. Hence, we only have a valid concentration for $\alpha \in (0, 0.133)$.

$\square$

**Proof of Theorem 6.10.** Let $\tau_0$ be the maximal epoch such that $B_\tau < \frac{200\log(16D|\mathcal{S}|\tau^2/\delta)}{\epsilon}$.

For all epoch $\tau \leq \tau_0$, the batch size is less than $2^{\tau_0}$. Since batch size doubles, until epoch $\tau_0$, we have the number of pulls for each arm $a \in [K]$ is less than $2 \cdot 2^{\tau_0} \leq 2\frac{200\log(16D|\mathcal{S}|\tau_0^2/\delta)}{\epsilon}$. Then the regret has to suffer $\frac{400\log(16D|\mathcal{S}|\tau_0^2/\delta)}{\epsilon}\Delta_a$ for each $a \in [K]$.

For $\tau > \tau_0$, $B_\tau \geq \frac{200\log(16D|\mathcal{S}|\tau^2/\delta)}{\epsilon}$. For each $a \in \mathcal{S}$, from Corollary 6.6, we have with probability at least $1 - \frac{\delta}{2|\mathcal{S}|\tau^2}$,

$$|\tilde{\mu}_a - \mu_a| \leq \beta_\tau.$$

Given an epoch $\tau > \tau_0$, we denote by $\mathcal{E}_\tau$ the event where for all $a \in \mathcal{S}$ it holds that $|\tilde{\mu}_a - \mu_a| \leq \beta_\tau$. and denote $\mathcal{E} = \cup_{\tau > \tau_0}\mathcal{E}_\tau$. By taking union bound, we have

$$\mathbb{P}(\mathcal{E}_\tau) \geq 1 - \frac{\delta}{2\tau^2},$$

and

$$\mathbb{P}(\mathcal{E}) \geq 1 - \frac{\delta}{2}\left(\sum_{\tau > \tau_0}\tau^{-2}\right) \geq 1 - \delta.$$

In the following, we condition on the good event $\mathcal{E}$. We first show that the optimal arm $a^*$ is never eliminated. For any epoch $\tau > \tau_0$, let $a_\tau = \arg\max_{a \in \mathcal{S}} \tilde{\mu}_a$. Since

$$(\tilde{\mu}_{a_\tau} - \tilde{\mu}_{a^*}) + \Delta_{a_\tau} = |(\tilde{\mu}_{a_\tau} - \tilde{\mu}_{a^*}) + \Delta_{a_\tau}| \leq |\tilde{\mu}_{a_\tau} - \mu_{a_\tau}| + |\tilde{\mu}_{a^*} - \mu_{a^*}| \leq 2\beta_\tau,$$

it is easy to see that the algorithm doesn't eliminate $a^*$.

Then, we show that at the end of epoch $\tau > \tau_0$, all arms such that $\Delta_a \geq 4\beta_\tau$ will be eliminated. To show this, we have that under good event $\mathcal{E}$,

$$\tilde{\mu}_a + \beta_\tau \leq \mu_a + 2\beta_\tau < \mu_{a^*} - 4\beta_\tau + 2\beta_\tau \leq \tilde{\mu}_{a^*} - \beta_\tau \leq \tilde{\mu}_{a_\tau} - \beta_\tau$$

which implies that arm $a$ will be eliminated by the rule. Thus, for each sub-optimal arm $a$, let $\tau(a)$ be the last epoch that arm $a$ is not eliminated. By the above result, we have

$$\Delta_a \leq 4\beta_{\tau(a)} = O\left(\sqrt{\frac{\log(|\mathcal{S}|\tau(a)^2/\delta)}{B_{\tau(a)}}} + \left(\frac{\log(|\mathcal{S}|\tau(a)^2/\delta)}{B_{\tau(a)}\epsilon}\right)^{1-\frac{1}{k}}\right).$$

Hence, we have

$$B_{\tau(a)} \leq O\left(\frac{\log(|\mathcal{S}|\tau(a)^2/\delta)}{\Delta_a^2} + \frac{\log(|\mathcal{S}|\tau(a)^2/\delta)}{\epsilon}\left(\frac{1}{\Delta_a}\right)^{\frac{k}{k-1}} + \frac{\log(D|\mathcal{S}|\tau_0^2/\delta)}{\epsilon}\right).$$

Since $|\mathcal{S}| \leq K$ and $2^\tau \leq T$ for any $\tau$. Thus,

$$B_{\tau(a)} \leq O\left(\frac{\log(K\log^2 T/\delta)}{\Delta_a^2}, \frac{\log(K\log^2 T/\delta)}{\epsilon}\left(\frac{1}{\Delta_a}\right)^{\frac{k}{k-1}}, \frac{\log(DK\log^2 T/\delta)}{\epsilon}\right),$$

Since the batch size doubles, we have $N_a(T) \leq 2B_{\tau(a)}$ for each sub-optimal arm $a$. Therefore, for all arm $a \in [K]$,

$$\mathcal{R}_T = \sum_{a\in[K]} N_a(T)\Delta_a \leq 2B_{\tau(a)}\Delta_a.$$

Let $\eta$ be a number in $(0, 1)$. For all arms $a \in [K]$ with $\Delta_a \leq \eta$, the regret incurred by pulling these arms is upper bounded by $T\eta$. For any arm $a \in [K]$ with $\Delta_a > \eta$, choose $\delta = \frac{1}{T}$ and assume $T \geq K$, then the expected regret incurred by pulling arm $a$ is upper bounded by

$$\mathbb{E}\left[\sum_{a\in[K],\Delta_a>\eta} \Delta_a N_a(T)\right] \leq \mathbb{P}(\bar{\mathcal{E}})\cdot T + O\left(\sum_{a\in[K],\Delta_a>\eta}\left\{\frac{\log T}{\Delta_a} + \frac{\log T}{\epsilon}\left(\frac{1}{\Delta_a}\right)^{\frac{1}{k-1}} + \frac{\log DT}{\epsilon}\Delta_a\right\}\right)$$

$$\leq O\left(\frac{K\log T}{\eta} + \frac{K\log T}{\epsilon\eta^{\frac{1}{k-1}}} + \frac{KD\log(DT)}{\epsilon}\right)$$

where the last term in the last inequality is based on following result: from the heavy-tailed assumption for rewards distributions in Definition 3.2, we have for any $a \in [K]$, $\mu_a \in [-D, D]$, so $\Delta_a = \mu^* - \mu_a \leq 2D$.

Thus the regret is at most

$$T\eta + O\left(\frac{K\log T}{\eta} + \frac{K\log T}{\epsilon\eta^{\frac{1}{k-1}}} + \frac{KD\log(DT)}{\epsilon}\right).$$

Taking $\eta = \max\left\{\sqrt{\frac{K\log T}{T}}, \left(\frac{K\log T}{T\epsilon}\right)^{\frac{k-1}{k}}\right\}$, the regret is at most

$$O\left(\sqrt{KT\log T} + \left(\frac{K\log T}{\epsilon}\right)^{\frac{k-1}{k}} T^{\frac{1}{k}} + \frac{DK\log(DT)}{\epsilon}\right).$$

For privacy guarantee, based on Laplacian mechanism in Definition 3.5, privacy guarantee for histogram learner in [49, Lemma 2.3], parallel composition theorem in Lemma B.2 and Post-processing in Lemma B.1, we can get the result.

$\square$

**Proof of Theorem 6.12.** The proof of the theorem is similar to the proof of Theorem 6.2, now the requirement for batch size to start to arm elimination becomes $\max\{\frac{\iota\log(16/\delta)}{\epsilon}, \frac{c_1\log(16D/\delta)}{\epsilon}, \frac{\log(16/\delta)}{\alpha_1^2}\}$ and the upper bound of $\Delta_a$ for each $a \in [K]$ is $2D$. Then we can get the result of upper bound for regret.

For privacy guarantee, based on Laplacian mechanism in Definition 3.5, privacy guarantee for histogram learner in [49, Lemma 2.3], parallel composition theorem in Lemma B.2 and Post-processing in Lemma B.1, we can get the result. □