# OpenReview forum: "On Private and Robust Bandits"
_NeurIPS.cc/2023/Conference — NeurIPS 2023 poster_

### Official Review · Reviewer_kQhg · 2023-06-16

**Soundness:** 3 good
**Presentation:** 4 excellent
**Contribution:** 3 good
**Rating:** 8
**Confidence:** 4

**Summary:**

In this paper, the authors study private and robust multi-armed bandits (MABs), where the heavy-tailed rewards are contaminated. They first present its minimax lower bound, characterizing the information-theoretic limit of regret with respect to privacy budget, contamination level, and heavy-tailedness. Then, they propose a meta-algorithm that builds on a private and robust mean estimation sub-routine PRM which could achieve nearly-optimal regrets. Finally, the authors run simulations to support their theoretical results.

**Strengths:**

1. The problem of private and robust bandits is well-motivated.
2. The upper bounds nearly match the lower bound, the paper is solid.
3. The writing is very clear, I enjoy reading the paper.
4. Connecting privacy with robustness using truncation is quite interesting.

**Weaknesses:**

My main question is about the last term in both Theorem 6.2 and Theorem 6.12. Intuitively, as the contamination converges to 0, the regret upper bound should be smaller, while the last term will converge to infinity. Could the authors discuss possible methods to get rid of the term or the difficulties?

**Questions:**

Please refer to the weaknesses above.

**Limitations:**

Yes.

---

> ### Author Rebuttal · Authors · 2023-08-08
>
> Thanks for your positive evaluation of our paper!
>
> Your observation is really sharp and points to a subtle part of Huber model.
>
> In fact, your observation is exactly the reason that we choose an upper bound $\alpha_1$ on the actual contamination level $\alpha$ and state the upper bound results in terms of $\alpha_1$ rather than $\alpha$. That is, for a very small but non-zero $\alpha$, one can choose a larger $\alpha_1$ to balance the regret.
>
> This subtle issue is also mentioned in one nice related work [1], see the remark after Theorem 7.4 and Remark 5.4.
>
>
> [1] Chen, Sitan, et al. "Online and distribution-free robustness: Regression and contextual bandits with Huber contamination." 2021 IEEE 62nd Annual Symposium on Foundations of Computer Science (FOCS). IEEE, 2022.

---

> > ### Comment · Reviewer_kQhg · 2023-08-15
> >
> > Thanks for your response. My concern is mostly addressed. I maintain my score and vote for acceptance.

---

### Official Review · Reviewer_Ssa1 · 2023-06-26

**Soundness:** 3 good
**Presentation:** 2 fair
**Contribution:** 3 good
**Rating:** 6
**Confidence:** 1

**Summary:**

The paper studies the MAB problem where the agent receives rewards that are both heavy tailed and contaminated according to Huber's process (provides true rewards w.p. $1-\alpha$ and provides samples from an arbitrary, unknown distribution w.p. $\alpha$). For the case of heavy-tailed rewards, both settings with finite $k$-th raw moment (the expected reward in a narrow range) and finite $k$-th central moment (expected deviation from the mean is in a relatively narrow range). The paper provides a regret lower bound for any algorithm in this setting, proposes a novel algorithms whose regret is analysed in both the finite $k$-th raw moment setting, as well as the finite central moment setting. Some experiments are provided in the Appendix though an existing algorithm, RUCB, appears to outperform the proposed algorithm.

**Strengths:**

The paper provides both a regret lower bound as well as nearly matching upper bounds for the regret of their proposed algorithm.

If contamination is removed from the setting, the regret upper bounds provided here match the lower bounds for the bounded central moment setting, filling a gap in the existing study of private bandits.

The algorithms are accompanied by experiments showcasing their performance against other existing algorithms designed for similar (but not identical) settings.

**Weaknesses:**

Presentation is lacking in my opinion and I found the paper hard to follow.

I am missing a thorough description of the problem statement describing the interaction between the agent and the environment in a self-contained, concise section. I found the quite hard to read as the setting isn't made explicit in a single place. I suggest this be done instead of the Preliminaries section (the definitions can be made in place as the need appears rather than having to keep track of them until they appear).

I would have liked to see a more explicit description of how privacy ties into this setting: whose privacy are we aiming to defend here and from whom? A more clear description of the setting should also clarify this, in my opinion.

 I find it hard to argue for the significance of the results here meeting the conference standards as the setting feels a bit too narrow. I would have liked to see the synergy between the two aspects studied here better articulated: is the combination of privacy and heavy tailed rewards in bandit settings more difficult than the sum of its parts? What does their interplay look like?

**Questions:**

See Weaknesses.

Also am I reading the experimental results correctly when assessing that the RUCB algorithm outperforms the one proposed here? Can you find a problem instance where your algorithm performs best among the baselines?

**Limitations:**

I do not see any potential negative societal impact arising from this work.

---

> ### Author Rebuttal · Authors · 2023-08-08
>
> Thanks for the suggestions on the presentation. We will address them as follows.
>
> - **Description of the problem setting** We have stated the MAB protocol in the first paragraph of the introduction section. As suggested by the reviewer,  we will re-emphasize it in the preliminary part (Section 3.1) in the next version.
>
> - **Description of privacy protection** We have explained privacy protection in lines 112-118. We will add a more intuitive explanation in the next version as follows: In other words, we protect the privacy of any individual user who interacts with the learning agent in the sense that an adversary observing the output of the learning agent (i.e., a sequence of actions) cannot infer too much about whether any particular individual has participated in this process, or the specific reward feedback of this individual.
>
> **Interplay of different parts** We first clarify that privacy and heavy-tailed rewards are not just the sum of the two parts. Rather, it introduces several interesting interplays. In particular, due to heavy-tailed rewards, it becomes difficult to guarantee privacy as the sensitivity is not bounded anymore. Thus, one cannot directly apply standard DP mechanisms, which assume a bounded sensitivity. Instead, one needs to first carefully control the sensitivity in some way while not leading to too much increase in regret. To this end, we employ the truncation method with a careful choice of truncation threshold to balance between privacy and regret. In addition to the interplay of privacy and heavy-tailed rewards, we also considered robustness in terms of contamination. Interestingly, we showed that truncation method can handle all three of them in a principled way.
>
> Given the recent popularity of research on privacy, heavy-tailed feedback, and robustness, we believe that our setting that studies the interplay of all three of them will have a broad audience. Practically speaking, our setting naturally covers many real-world scenarios. Theoretically speaking, our paper not only provides some useful tools for the field (e.g., optimal concentration result), but reflects the recent trends on the interplay of privacy and robustness [1]-[3].
>
> [1] Samuel B Hopkins, Gautam Kamath, Mahbod Majid, and Shyam Narayanan. Robustness implies privacy in statistical estimation. arXiv preprint arXiv:2212.05015, 2022.
>
> [2] Hilal Asi, Jonathan Ullman, and Lydia Zakynthinou. From robustness to privacy and back. arXiv preprint arXiv:2302.01855, 2023.
>
> [3] Kristian Georgiev and Samuel B Hopkins. Privacy induces robustness: Information-computation gaps and sparse mean estimation. arXiv preprint arXiv:2211.00724, 2022.
>
> **Experimental results** We first clarify that we design the experiments by choosing DPRSE as the private bandit benchmark and RUCB as the robust bandit benchmark. RUCB is only a robust but not private algorithm and hence it's better than ours in terms of regret. Our algorithm not only handles robustness but also guarantees differential privacy, which naturally leads to a worse regret guarantee compared to RUCB.

---

> > ### Comment · Reviewer_Ssa1 · 2023-08-13
> > **Further clarifications on the setting**
> >
> > Thank you for your response. I found your clarifications regarding the interplay between heavy-tailed rewards and the standard DP mechanisms and those surrounding the experimental results to be very informative.
> >
> > I believe the description provided in the sections you mention and the description in the response does not provide sufficient clarity about the setting to someone not intimately familiar with the privacy side of MABs (such as myself). Please allow me to formulate what I am finding unclear more explicitly:
> >
> > - What information does the adversary have access to (entire history of actions of the decision maker? The entire history of realised rewards as well? What does the adversary know about the users whose privacy we aim to protect?) and what is the information we are aiming to keep private (whether or not a user presented themselves to the system?)? Presenting this aspect of the setting together with the MAB protocol would make the paper a lot more accessible, in my opinion, as it crystallises the problem in one place and allows the reader to easier recognise the use of concepts introduced later in the paper.
> >
> > - "Practically speaking, our setting naturally covers many real-world scenarios." - Can you provide an example scenario where this setting applies: heavy-tail rewards and a setting where it is "easy" (possible) for an adversary to infer the information we aim to keep private (as described in the answer to the point above) when no privacy preserving mechanisms are employed? Such an example would make the significance of the problem a lot more obvious to me.

---

> > > ### Author Response · Authors · 2023-08-14
> > > **Further rebuttal to Reviewer Ssa1**
> > >
> > > We thank the reviewer for the follow-up. We are glad to hear that you find our clarifications on the interplay between heavy-tailed rewards and privacy very informative.
> > >
> > > We would like to provide further clarifications on the setting by answering your specific questions.
> > >
> > > **Privacy notion in the paper** We adopt the central DP for MABs as our privacy notion, which has been widely used in previous works on private MABs [19,20,24]. We would like to give more details about this notion in the following two steps.
> > >
> > > 1. We first give the standard interpretation of this privacy notion. In particular, the adversary is an external party that has the information on all the $T$ actions during the MAB learning process. The information we aim to protect is the reward generated by each of the $T$ users. Central DP (cf. Def. 3.4) protects user's reward in the following sense: The external third-party (adversary) cannot determine the reward of any user $t \in [T]$ with high confidence by observing all $T$ generated actions. This is because, by defintion, while changing the reward of any user $t$, the output action sequences are indistinguishable in probability.
> > >
> > > 2. In fact, in addition to the above standard interpretation, central DP also offers the following stronger protection: The adversarially can be all other  $T-1$ malicious users and even if they can collude adversarlly to induce the learning agent to reveal information about the reward of the remaining user, they cannot infer too much about the reward of the remaining user.
> > > Further, if one considers replacing the reward at any $t$ to a special symbol to represent the event of removal of the corresponding user $t$ in the input sequences, then central DP also protects the information whether one user has participated in the learning process or not.
> > >
> > >
> > > **A Concrete Example** We will use **dynamic pricing** as a concrete example scenario where the reward can be heavy-tailed and there exists a privacy leakage of the reward if no privacy protection is adopted.
> > >
> > >  *Scenario:*  Online Retailer Selling Sensitive Products. The MAB learning agent sequentially chooses a action (price for the product) based on previous reward feedback (demand) so as to maximize the total expected revenue.
> > >
> > >  *Heavy-tailed Demand:* The demand for the product may exhibit a heavy-tailed pattern due to factors such as:
> > >
> > >  - Seasonal outbreaks leading to sudden spikes in demand.
> > >  - Public awareness campaigns or celebrity endorsements causing immediate interest.
> > >  - Regulatory changes making the product more accessible to a broader population.
> > >
> > > *Privacy leakage:* Suppose the product is a specific medication used to treat a highly sensitive or stigmatized health condition. Thus, the particular demand of a user (the reward in MAB formulation) is highly sensitive. As discussed in [11, 12], an adversary might place orders immediately before and after a person of interest (i.e., target user) and if he sees a slight spike in his received prices, he might be able to infer the purchase decision (demand/reward) of the target user.
> > >
> > > Please let us know if our clarifications help to resolve your concern and we are happy to engage more if there are any additional questions.

---

> > > > ### Comment · Reviewer_Ssa1 · 2023-08-14
> > > > **Thank you for the reply**
> > > >
> > > > Thank you for clarifying further! I believe this eases a lot of my concerns and will adjust my score appropriately.

---

### Official Review · Reviewer_nGB6 · 2023-07-06

**Soundness:** 3 good
**Presentation:** 4 excellent
**Contribution:** 3 good
**Rating:** 7
**Confidence:** 3

**Summary:**

This work studies the specific setting of heavy-tailed bandits with Huber contamination and differential privacy constraints. They first give a regret lower bound, tightly characterizing the minimax rate in terms of all the parameters involved in this setting. Then, they provide matching (up to log terms) upper bounds. The crucial technical novelty is to use new concentration bounds they developed for private and robust estimation via truncation methods in bandits.

**Strengths:**

They derive the first minimax rates in this robust+private bandit setting and give matching upper bounds. The paper is also generally well-written and has many useful remarks comparing the results and techniques with previous works. The algorithmic ideas and intuition behind the new estimation scheme were easy to follow.

**Weaknesses:**

* I think some discussion on problem-dependent rates in this setting would be interesting. The paper could at least comment on why their analyses don't extend easily to obtain problem-dependent rates or what the minimax problem-dependent rates might look like in terms of $\epsilon, k,\alpha$.
* There are many recent works on "bandits with total corruption budget" (e.g., the cited paper Lykouris et al., 2018). It would be good to include some comparison with this setting, even if just in terms of experiments. For instance, the $T\cdot \alpha^{1-1/k}$ term in this paper's minimax regret rate seems to be similar to the additive corruption term sometimes seen in this other setting (Theorem 1; Gupta et al., COLT 2019). Can the analyses in this paper extend to this setting?

**Questions:**

See above.

**Limitations:**

No forseeable negative societal impacts.

---

> ### Author Rebuttal · Authors · 2023-08-08
>
> We thank the reviewer for the positive evaluation of our paper! We would like to provide the following clarifications to the reviewer's questions.
>
> **Minimax vs. problem-dependent bound**  Thanks for your sharp comments. Let us approach your question in the following steps.
>
> - Problem-dependent upper bound:  Our current analysis can also yield problem-dependent upper bounds. In particular, by line 586-587, if $\alpha \leq c \Delta_{\min}$ (where $\Delta_{\min}$ is the minimal gap; $c$
>  is some constant), then one can determine the number of pulls for each sub-optimal arm, hence a standard problem-dependent bound, which mimics the one established for the case of Gaussian inlier rewards [1].
>
> - Problem-dependent lower bound: it is unclear for private and robust bandits, but some work has been done for corrupted heavy-tailed bandits; see Theorem 1 in [2]. One interesting future direction is to study how to leverage both the insights in [2] and the lower bound under privacy to derive a problem-dependent lower bound for private and robust bandtis.
>
> [1] Sayash Kapoor, Kumar Kshitij Patel, and Purushottam Kar. Corruption-tolerant bandit learning, Machine Learning 108.4 (2019): 687-715.
>
> [2] Debabrota Basu, Odalric-Ambrym Maillard, and Timothée Mathieu. Bandits corrupted by nature: Lower bounds on regret and robust optimistic algorithm. arXiv preprint arXiv:2203.03186,
> 405 2022.
>
> **Comparision with "bandits with total corruption budget"**
>
> - First, we clarify that the "bandits with total corruption budget" and "Huber contamination model" is different in nature, as they are models for different contamination scenarios. This is also reflected in Section 3.4 of [1].
>
> -  At a high level, Algorithm 1 in [2] is similar to our meta-algorithm in the spirit of arm elimination, forgetting, and batching since both algorithms proceed in epochs that increase exponentially in length and only use the most recent epoch to calculate statistics. The difference is that Algorithm 1 in [2] chooses the arm with a probability at step 8 and never completely eliminates any arm, but our algorithm removes the arm based on the confidence radius.
> - One possible approach to further provide a privacy guarantee to Algorithm 1 in [2] is as follows: As in our paper,  one can also add some noise on the average of rewards at step 9. Then there should be an additional term related to noise in concentration inequality (1) in [2]. This approach is reasonable because Algorithm 1 in [2] also enjoys doubling and forgetting. We believe that one can use this approach to derive a private version regret guarantee of Algorithm 1 in [2].
>
> [1] Niss, Laura, and Ambuj Tewari. "What You See May Not Be What You Get: UCB Bandit Algorithms Robust to $\varepsilon $-Contamination." In Conference on Uncertainty in Artificial Intelligence, pp. 450-459. PMLR, 2020.
>
> [2] Gupta, Anupam, Tomer Koren, and Kunal Talwar. "Better algorithms for stochastic bandits with adversarial corruptions." In Conference on Learning Theory, pp. 1562-1578. PMLR, 2019.

---

> > ### Comment · Reviewer_nGB6 · 2023-08-13
> > **Thanks**
> >
> > Thank you for the detailed and clarifying response. My concerns were mostly writing/discussion suggestions, and I'm still in support of accepting the paper.

---

### Official Review · Reviewer_xHN8 · 2023-07-10

**Soundness:** 2 fair
**Presentation:** 3 good
**Contribution:** 2 fair
**Rating:** 6
**Confidence:** 3

**Summary:**

The paper studies the private and robust multi-armed bandits, where the rewards are heavy-tailed or contaminated. It proposes a meta-algorithm which is based on private and robust mean esitmation sub-routine incorporating reward truncation and Laplace mechanism.

**Strengths:**

The paper establish the minimax regret lower bound for private and robust MABs.

It proposes a meta-algorithm matching the lower bound.

It takes into account both reward contamination and heavy-tailed cases.

The analyses are relatively complete.

**Weaknesses:**

The current setting seems to require the horizon, T, to be known in advance. If T is unknown or infinity, how does it change the differential privacy definition and the technical difficulty.

The paper needs to include additional literature review about heavy-tailed cases, especially in the line of Catoni's estimator (with application in bandits). For example,

1. Olivier Catoni. Challenging the empirical mean and empirical variance: a deviation study
2. Gabor Lugosi and Shahar Mendelson. Mean estimation and regression under heavy-tailed distributions: A survey.
3. Sujay Bhatt, Guanhua Fang, Ping Li, Gennady Samorodnitsky. Nearly Optimal Catoni’s M-estimator for Infinite Variance

Some detailed explanations of differences between proposed method and DPRSE should be given.

In supplementary, it seems that Lemma B.2 is never used. According to the current algorithm, the pararell composition should be sufficient?

**Questions:**

Overall, I think the paper is well organized and easy to follow. See questions in weakness part.

**Limitations:**

No.

---

> ### Author Rebuttal · Authors · 2023-08-08
>
> Thanks for your time and comments. We are glad you find our analyses are relatively complete. We will recap your comments and present our detailed response. We hope our answers will resolve your concern.
>
> **Unkown and infinity horizon T**
> - We first clarify when $T$ is unknown and infinity, the differential privacy definition is the same for differentially private online learning under the event-level privacy framework of [1]. The definition is the same under two cases also has been discussed in Section 2.2 of [2].
> - When $T$ is unknown and infinity, in fact, once armed with our novel PRM modules, one can also use other exploration strategies like UCB in [3] for anytime regret guarantee. One difference is that now instead of first pulling each arm once, it needs to pull each arm $\mathcal{T}$ times to ensure that concentration kicks in later. This is in fact, not surprising since on the high level, the analysis of SE and UCB is very similar, i.e., doubling and forgetting.
>
> - We also note that instead of UCB, one can also adapt it to the Thompson sampling strategy, e.g., [4], with our PRM module. Again, the key idea is doubling and forgetting.
>
>
> We will include the above discussion in the next version to highlight the flexibility of our PRM modules.
>
> [1] Dwork, C., Naor, M., Pitassi, T., and Rothblum, G. N. Differential privacy under continual observation. In Proceedings of
> the forty-second ACM symposium on Theory of computing, pp. 715–724, 2010.
>
> [2] Hu, Bingshan, Zhiming Huang, and Nishant A. Mehta. "Optimal algorithms for private online learning in a stochastic environment." arXiv preprint arXiv:2102.07929 (2021).
>
> [3] Azize, Achraf, and Debabrota Basu. "When privacy meets partial information: A refined analysis of differentially private bandits." *Advances in Neural Information Processing Systems* 35 (2022): 32199-32210.
>
> [4] Hu, Bingshan, and Nidhi Hegde. "Near-optimal Thompson sampling-based algorithms for differentially private stochastic bandits." In Uncertainty in Artificial Intelligence, pp. 844-852. PMLR, 2022.
>
> **Additional literature review about heavy-tailed cases** Thanks for pointing out these important and nice works. We will include them in the next version.
>
> **Difference between our proposed algorithms and DPRSE**
> - First, DPRSE in [5] is only designed for handling privacy and heavy-tailed rewards, i.e., no robustnees with respect to contamination. In contrast, our proposed algorithm can handle privacy, heavy-tailed rewards and Huber contamination altogether.
> - Second, even if one only considers privacy and heavy-tailed rewards, our algorithm can handle the important case where the central moment is bounded while DPRSE cannot. This is because DPRSE is desinged only for the finite raw moment case.
>
> [5] Youming Tao, Yulian Wu, Peng Zhao, and Di Wang. Optimal rates of (locally) differentially private heavy-tailed multi-armed bandits. arXiv preprint arXiv:2106.02575, 2021
>
> **Parallel composition** Yes, we didn't use Lemma B.2 and parallel composition is sufficient. Thanks for pointing out this and we will remove it in the next version.

---

### Decision · Program_Chairs · 2023-09-21

**Decision:**

Accept (poster)

**Comment:**

The reviewers and myself have recognized that the many strong contributions of these papers qualify it to be accepted to Neurips 2023.

The paper is clear and well-written but one of the reviewers suggests that the discussion on the meaning of “privacy” in this paper should be added to the final version.